

# Western Indian Ocean bottom water temperature calibration – are benthic foraminifera Mg/Ca ratios a reliable palaeothermometry proxy?

Viktoria Larsson[1] and Simon Jung[1]

[1] School of Geosciences, University of Edinburgh, Edinburgh, EH9 3FE, United Kingdom

*Correspondence to:* Viktoria Larsson (viktorialarsson3@gmail.com)

**Abstract:** Mg/Ca ratios measured in benthic foraminifera have been explored as a potential palaeothermometry proxy for
bottom water temperatures (BWT). Mg/Ca-BWT calibrations from the Indian Ocean are rare and comprise conflicting results.
Inconsistencies between studies suggest that calibrations may need to be region specific. The aim of this study was to develop
benthic foraminifera (*Uvigerina peregrina, Cibicidoides wuellerstorfi* and *Cibicidoides mundulus*) based Mg/Ca – BWT
calibrations in the tropical western Indian Ocean. Testing variations of existing analytical protocols, aimed at optimising
cleaning of the foraminifera while avoiding sample loss in the process, entailed that a previously established protocol by Barker
et al. (2003) was the most suitable for our study. The majority of samples of *Cibicidoides mundulus* and *Uvigerina peregrina*,
however, remained contaminated, rendering those data unusable for Mg/Ca core-top calibrations. Only Mg/Ca ratios in
*Cibicidoides wuellerstorfi* - BWT calibration with the relationship being: $Mg/Ca = 0.19 \pm 0.02 *$
$BWT + 1.07 \pm 0.03, r^2 = 0.87$. While this result differs to some degree from previous studies it principally suggests that
existing core-top calibrations from the wider Indian Ocean can be applied to core-tops in the western Indian Ocean. The
agreement of Mg/Ca ratios at lower temperatures in *Cibicidoides wuellerstorfi*, *Cibicidoides mundulus* and *Uvigerina peregrina*
with Mg/Ca ratios reported for these species at low temperatures in other studies supports this conclusion. Many uncertainties
surrounding the Mg/Ca proxy exist and more calibration studies are required to improve this method.



## 1.    Introduction

The global thermohaline circulation is crucial for distributing heat, nutrients, oxygen and salinity and it partially controls the oceanic carbon uptake (Blunier et al., 1998; Clark et al., 2002).  Specifically, the re-/distribution of heat is an important driver of climate change, with Antarctic Intermediate Water (AAIW), Antarctic Bottom Water (AABW) and North Atlantic Deep Water (NADW) representing crucial water masses. Palaeoceanographic reconstructions have greatly improved our understanding of the sensitivity and of changes of the thermohaline circulation. On glacial-interglacial time scales for example NADW and AABW importance seems to have alternated between NADW being more prominent during interglacials and AABW during glacials (Duplessy et al, 1988, Curry et al., 1988, Sarnthein et al., 1994 ). These water mass reorganisations had largescale implications on global climate (Blunier et al., 1998).

There are a range of proxies measured in foraminifera used to reconstruct changes in seawater properties through time. Stable oxygen isotopes ($\delta^{18}$O) have been widely applied to identify changes in water column properties (Kroopnick, 1985; Lynch-Stieglitz and Fairbanks, 1994). Straightforward interpretation, however, is hampered due to stable oxygen isotopes reflecting more than one environmental factor, i.e. ambient temperatures and seawater $\delta^{18}$O with the latter being controlled by global ice volume and evaporation-precipitation in the water mass source region (Emiliani, 1955; Shackleton, 1974). In order to improve the use of stable oxygen isotopes, independent temperature proxies have been developed (Elderfield and Ganssen, 2000; Lea et al., 1999; Nuernberg, 1995; Nuernberg et al., 1996). Mg/Ca based temperature estimates in planktic foraminifera for example are widely used as a proxy for sea surface temperature (SST, Barker, 2005). The use of Mg/Ca ratios in benthic foraminifera for reconstructions of bottom water temperature (BWT) is being explored (Rosenthal et al. 1997; Elderfield et al., 2006) although a uniformly adopted method is still being developed.

### 1.1.    Mg/Ca ratios - a proxy for temperature

During the formation of tests of foraminifera, divalent ions of trace elements such as $Mg^{2+}$ are also incorporated in the calcite lattice (Erez, 2003). Resulting Mg/Ca ratios in benthic foraminifera depend on the Mg/Ca ratio of ambient seawater and elemental partitioning during calcite precipitation (Elderfield et al., 1996; Gussone et al. 2016). On glacial-interglacial timescales Mg/Ca ratios in seawater can be considered constant due to long residence times for Ca and Mg (~$10^6$ and $10^7$ years, respectively). Existing core-top calibrations show a positive correlation between Mg/Ca ratios in benthic hyaline low magnesium calcite foraminifera and modern BWT (Martin and Lea, 2002; Elmore et al., 2015). Temperature appears to be the dominant environmental factor in some species of *Cibicidoides spp.* (Rosenthal et al., 1997) but other factors including carbonate ion saturation might also have an effect (Elderfield et al., 2006; Yu and Elderfield, 2008). There is discussion on the importance of factors controlling Mg/Ca incorporation in benthic foraminifera, with e.g. Yu and Elderfield (2008) suggesting carbonate ion saturation being dominant whereas Lear et al.'s (2002) work implies only a minor influence. There is also evidence suggesting that the Mg/Ca - temperature relationship and other controlling factors including carbonate ion effect are spatially varying (Bryan and Marchitto, 2008).

### 1.2.    Mg/Ca analysis – a brief summary

The chemical cleaning procedure is a critical step essential for accurate determination of Mg/Ca ratios in foraminifera (Barker et al., 2003) due to the generally low Mg/Ca concentration ratios, entailing the need to remove Mg containing contaminants (Barker et al., 2003; Marr et al., 2013). Concurrently, carbonate dissolution of tests may affect Mg/Ca ratios (Lear et al., 2002) and therefore the aim of a cleaning procedure is to effectively clean tests while minimising sample loss (Barker et al., 2003). Silicate contamination is the most critical contaminant affecting Mg/Ca ratios, followed by Mn-oxide coatings (Barker et al.,



2003). The two most widely used cleaning methods are the 'Mg cleaning method' also referred to as the 'oxidative cleaning method' by Barker et al. (2003) based on Boyle and Keigwin (1985), and the 'Cd cleaning method' also referred to as the 'reductive oxidative cleaning method' by Boyle and Keigwin (1985) and Rosenthal et al. (1995, 1997b). Both methods include successive rinses with ultrapure water followed by methanol cleaning, and an oxidative cleaning step to remove silicates. The

'Cd cleaning method' in addition includes a reductive step to remove Mn-oxide coatings. The procedure was originally intended for determination of Cd/Ca ratios (Boyle and Keigwin, 1985) because Cd concentrations in calcite are significantly lower than Mg concentrations and therefore contamination is more critical (Marr et al., 2013). While the more aggressive 'Cd cleaning procedure' is not viewed as needed for accurate Mg/Ca analyses (Barker et al., 2003; Yu et al., 2008), it is still used (Stirpe et al., 2021) amid continued uncertainty surrounding the requirement of additional rigour for accurate Mg/Ca analyses (e.g. Pena

et al., 2005; Hazenfratz et al., 2017). On the other hand, the additional reductive step seems to lower Mg/Ca ratios due to partial preferential dissolution of Mg-rich calcite (Barker et al., 2003; Yu et al., 2007) causing a significantly larger (~15%) lowering in Mg/Ca ratios than the increase in Mg/Ca ratios due to diagenetic coatings (only ~1%; Barker et al. (2003) and Yu et al. (2007)). Most studies using Mg/Ca ratios in benthic foraminifera have utilised the 'Mg cleaning procedure' (e.g. Elderfield et al., 2006; Elderfield et al., 2010, Elmore et al., 2015), thereby targeting the most important contaminants, i.e. silicate

contamination, organic matter, Mn-oxide coatings, and secondary calcification (Barker et al., 2003). Furthermore, rather than analysing multiple whole-shell specimen of foraminifera for analysis by solution in ICP-MS/ICP-OES, Stirpe et al. (2021) used laser ablation ICP-MS measuring Mg/Ca ratios revealing unevenly distributed Mg/Ca ratios between different chambers in *Uvigerina spp.* Also, Branson et al. (2013) analysed tests of two species of planktic foraminifera showing a systematic banding of Mg distribution. Based on these findings, whole-shell analysis by solution remains the most appropriate method of

determining calcite Mg/Ca ratios (Stirpe et al., 2021).

### 1.3. Species specific Mg/Ca – temperature calibrations

Earlier studies developed Mg/Ca – BWT calibrations using mixed benthic foraminifera of the same genera, mostly *Cibicidoides spp.* (e.g. Rosenthal et al., 1997). Later work, however, implies a species-specific temperature sensitivity driving the Mg/Ca signal (Lear et al., 2002; see figure 1 for locations within the Indian Ocean). *Cibicidoides wuellerstorfi* have been one of the

most widely used benthic species for stable $\delta^{18}$O and $\delta^{13}$C reconstructions. It is advantageous over other benthic species because it is a true epifaunal species and suggested to record bottom water properties (Lutze and Thiel, 1989). However, some core-top studies suggest Mg/Ca incorporation in *Cibicidoides wuellerstorfi* is significantly influenced by carbonate ion saturation (Elderfield et al., 2006 - see figure 1 for locations within the Indian Ocean; Yu and Elderfield, 2008), limiting its use as a proxy for BWT. In contrast Mg/Ca ratios in shallow endofaunal *Uvigerina spp.* seem to be independent of carbonate ion saturation

(Yu and Elderfield, 2008; Elderfield et al., 2010; Stirpe et al., 2021; Elmore et al., 2015) with Mg/Ca ratios in *Uvigerina peregrina* being useable as a proxy for temperature at intermediate depths <2.4 km, based on calibration data from the southwest Pacific Ocean (Stirpe et al., 2021). Uncertainties remain (Stirpe et al., 2021), however, entailing the need for core-top calibrations in various areas to assess the robustness of the Mg/Ca thermometry in benthic foraminifera.

In order to help improving our understanding of Mg/Ca based thermometry in deep/intermediate water based on benthic

foraminifera, we present benthic foraminiferal (*Uvigerina peregrina*, *Cibicidoides wuellerstorfi* and *Cibicidoides mundulus*) based Mg/Ca –temperature calibrations using core top samples from the western tropical Indian Ocean and compare those with calibrations from the Indian Ocean. We also assess the usefulness of adaptions to cleaning procedures.



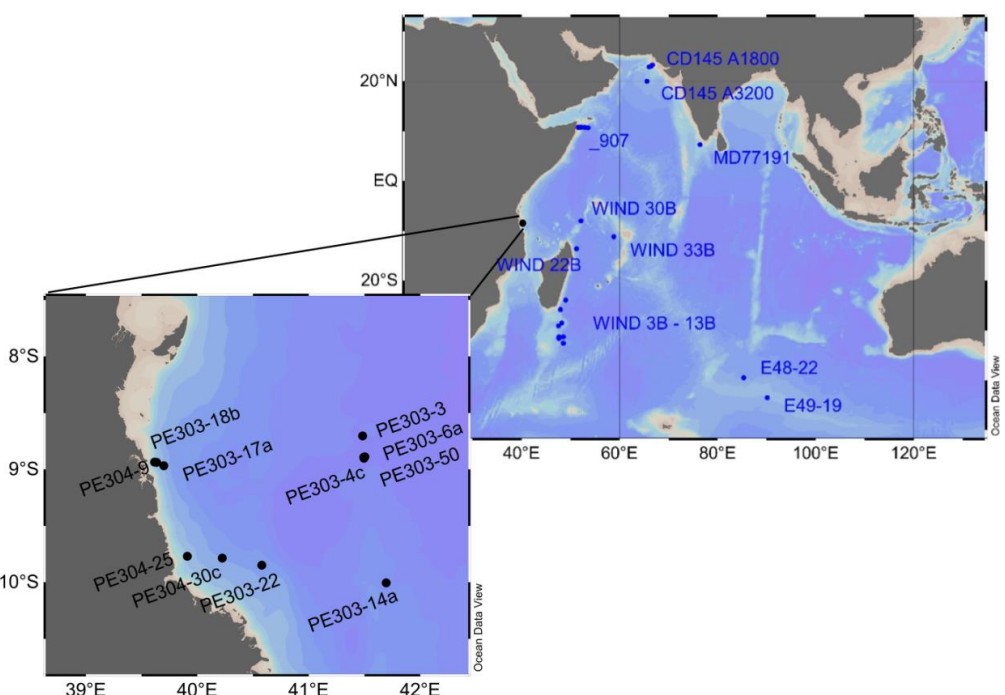

**Fig. 1** Map showing location of existing sediment core-top benthic Mg/Ca – BWT calibrations in the Indian Ocean (blue) and the location of cores analysed in this study (black) *Map produced in Ocean Data View.*

## 2. METHODOLOGY

### 2.1. Sample location and hydrography

This study is based on a transect of sediment surface samples retrieved from 370 to 3400 m water depth off Tanzania (see Figure 1 and Table 1). In order to optimise sample quality, only box core or multicorer samples were used (Table 1). The cores have been taken during the "Indian – Atlantic Exchange (INATEX)" (Brummer and Jung, 2009) and the "Tropical Temperature History during Paleogene Global Warming Events (GLOW)" (Kroon et al., 2010) expeditions. In the modern western Indian Ocean, the water column at our core-top transect comprises three bottom to intermediate water masses, i.e. AABW below 4000 m, Circumpolar Deep Water (CDW) between 2000 and ~3500 m (Figure 3, You et al., 2000; McCave et al., 2005). Above CDW there is a zone influenced by Red Sea Water (RSW) and/or Antarctic Intermediate Water (AAIW), with both water masses extending south- and northwards, respectively, controlling intermediate depth water properties at the location of our core top transect (Figure 2-3; sensu Talley, 1999; Gründlingh, 1985 and McCave et al., 2005). Based on a nearby CTD profile, bottom water temperatures at the transect range from 1 to 10˚C (Table 1).




**Tab. 1.** Details of core-top samples of benthic foraminifera analysed in this study

| Core | Core type | Latitude (°W) | Longitude (°E) | Depth (m) | BWT (°C) | Species | Size fraction (µm) | Specimens (count) |
|---|---|---|---|---|---|---|---|---|
| PE303-3 | BC | -8.7034 | 41.48307 | 3006 | 1.74[5] | *Cib. wuellerstorf* | 250 – 450 | 15 |
|  |  |  |  |  |  | *Cib. spp.* | 250 – 450 | 9 |
| PE303-4c | BC | -8.89300 | 41.49298 | 3179 | 1.61[5] | *Uvi.spp.* | 150 – 450 | 8 |
|  |  |  |  |  |  | *Cib. wuellerstorfi* | 150 – 450 | 7 |
| PE303-5 | BC | -8.90167 | 41.49507 | 3371 | 1.46[5] | *Cib. spp.* | 150 - 450 | 12 |
| PE303-6a | BC | -8.88828 | 41.5038 | 3323 | 1.48[5] | *Uvi. spp.* | 150 – 250 | 15 |
|  |  |  |  |  |  | *Cib. mundulus* | 250 – 450 | 5 |
|  |  |  |  |  |  | *Cib. wuellerstorfi* | 150 – 450 | 19 |
| PE303-14a | BC | -10.00415 | 41.69455 | 2560 | 2.32[2] | *Uvi. spp.* | 250 – >450 | 6 |
|  |  |  |  |  |  | *Cib. spp* | 250 – 450 | 10 |
| PE303-17a | BC | -8.96737 | 39.70033 | 1105 | 5.34[5] | *Uvi. spp.* | 250 – 450 | 18 |
| PE303-18b | BC | -8.93778 | 39.63465 | 490 | 8.59[5] | *Uvi. spp.* | 250 – 450 | 44 |
|  |  |  |  |  |  | *Cib. Wuellerstorfi* | 250-450 | 8 |
|  |  |  |  |  |  | *Cib. mundulus* | >450 | 4 |
| PE303-22 | BC | -9.84817 | 40.57933 | 1855 | 2.95[5] | *Cib. wuellerstorfi* | 150-250 | 15 |
| PE304-9 | MC | -8.93555 | 39.61638 | 370 | 9.91[5] | *Uvi. spp.* | 150 – 450 | 15 |
|  |  |  |  |  |  | *Cib. spp* | 150 – 450 | 19 |
| PE304-25 | MC | -9.76978 | 39.91057 | 482 | 8.68[5] | *Uvi. spp.* | 250 - >450 | 6 |
| PE304-30c | MC | -9.78565 | 40.22365 | 1471 | 4.29[5] | *Cib. spp.* | 250 – 450 | 7 |

BC = box core, MC = multicore

[2,5] Bottom water temperatures from nearest CTD profiles from Birch et al. (2013): [2] = GLOW Station 2 and [5] = GLOW Station 5.




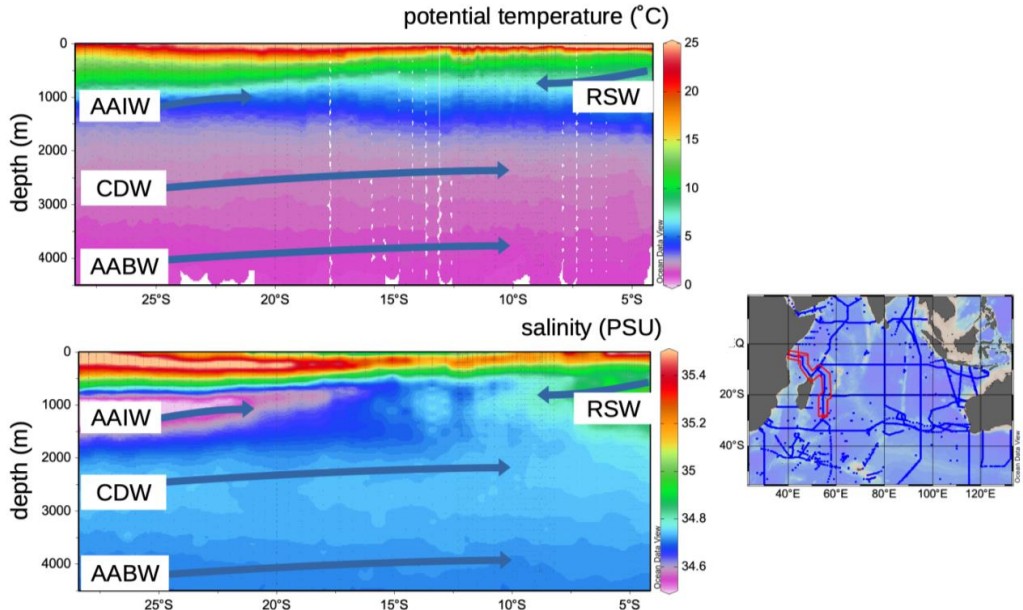

**Fig. 2.** North-south cross section of temperature and salinity over depths 0-4500 m in the western Indian Ocean from 28˚S to 4˚S. Arrows show flow direction of the main water masses. Temperature and Salinity data from GLODAPv2 (Lauvset et al., 2022). Map produced in Ocean Data View.

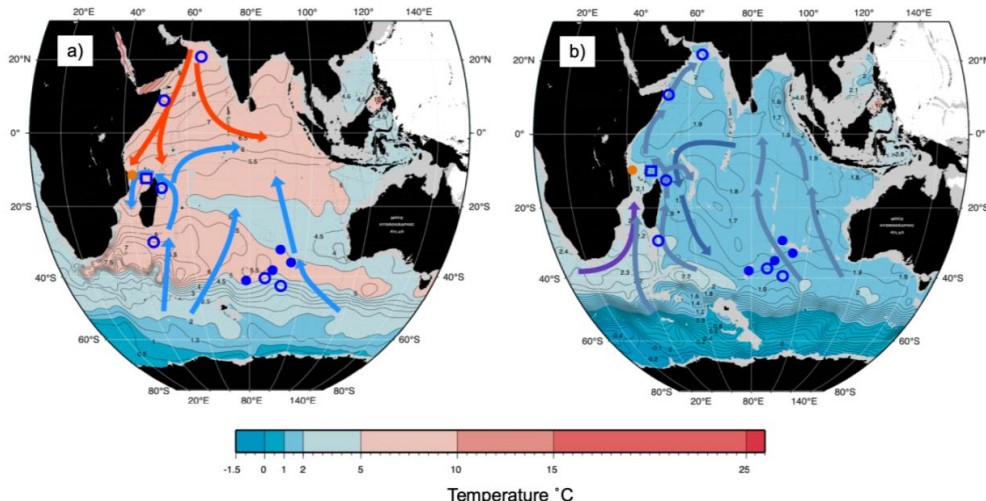


**Fig. 3.** Map of potential temperature adapted from The WOCE Indian Ocean Atlas (Talley et al., 2013) at intermediate and deep-water depths. a) 1000 m with general intermediate water circulation; blue: AAIW and red: RSW b) 2500 m with general deep water circulation; lighter blue: LCDW, darker blue: NIDW/UCDW and purple: NADW. Approximate location of core-tops from the Indian Ocean; orange circle: this study, open square: nearest CTD transect, filled circle: Healey et al. (2008) and

open circle: Elderfield et al. (2010).





## 2.2. Core top sample preparation

Mg/Ca can be affected by intra-shell, intra-species, and inter-species variability referred to as 'vital effects' (Bentov and Erez, 2006; de Noijer., 2014; and discussions therein). In order to minimize these effects, we have focussed, (where possible), on

analysing multiple whole specimen of three benthic foraminifera species; *Uvigerina peregrina*, *Cibicidoides wuellerstorfi* and *Cibicidoides mundulus.*. In most cases these were picked from comparatively small size fraction windows (i.e. 250-355 µm and 250-355 µm). Only in a small number of cases a wider size fraction window was used due to low foraminifera abundances (Table 1). Specimens with no signs of stains, discoloration, fragmentation or post depositional calcification were selected based on previous studies suggesting post depositional effects on Mg/Ca ratios (Lear et al., 2002) (see qualitative observations

in Appendix C). Because temperature sensitivity of Mg/Ca in *Cibicidoides spp.* might be species specific (Lea et al., 1999; Gussone et al., 2016), species of *Cibicidoides spp.* were analysed separately except for samples with abundance <5 specimens (Table 1). Sample PE303-18b was split into two to check for intra sample variation (the only sample with sufficient sample size). In order to test the cleaning procedure specimens from the planktic foraminifera *G. ruber* were picked from the 250-355 µm size fraction of samples from core NIOP929 because there were insufficient planktic and benthic foraminifera specimens

in our off Tanzania core top transect to carry out these tests. The samples have been wet sieved over a >63µm screen and dried at 40°C.

All samples were chemically cleaned using water and methanol rinses to remove silicates, hydrogen peroxide treatment to remove organic matter followed by an acid rinse to remove residual treatment chemicals (Barker et al., 2003). The rigour needed to sufficiently clean samples depend on a number of factors including sediment composition and foraminifera

morphology, i.e. some species trapping contaminants more than others. Therefore we have adapted the Mg cleaning methodology by Barker et al., (2003) to find the appropriate level of cleaning required (optimum removal of contaminants while minimising sample loss) for the benthic samples analysed in this study.

In the first experiment, the cleaning protocol of Barker et al., (2003) was followed except for reducing the time during methanol washes (25 seconds repeated thrice compared to 1-2 min repeated once) following previously analysed samples in the

laboratory. The chemicals were prepared following Barker et al., (2003). Traditionally, the procedure involves crushing of foraminifera between two glass plates. Given the small sample volume in our study, we tested individual crushing of foraminifera specimens using a metal pin and glass mortar to open the test chambers. The samples were transferred to acid-cleaned Eppendorf tubes with 500 µl MilliQ water and washed with MilliQ followed by methanol, both using an ultrasonic bath. This was followed by a hydrogen peroxide treatment in a hot water bath and an acid leach using nitric acid (see protocol

in Barker et al., 2003).

The results of the first experiment, with some variability, show that average Ca concentrations (normalised to the number of tests) of samples crushed using two glass plates were lower (5.81 ppm) than in samples crushed using a metal pin and glass mortar (9.53 ppm), see Table A1 and A2 in Appendix A, suggesting less sample loss in the latter. The average Mg/Ca ratios of samples crushed using two glass plates and using metal pin and glass mortar was similar, i.e. 3.43 mmol mol$^{-1}$ and 3.53 mmol

mol$^{-1}$, respectively (see Table A1 and A2 in Appendix A), suggesting that crushing using a metal pin and glass mortar does not introduce more Mg or Ca bearing contaminants than the technique using two glass plates. Fe and Al concentrations were below the limit of detection in all samples (<0.0070 and <0.0079 ppm) suggesting no silicate contamination. Because the technique using a metal pin and glass mortar entailed less sample loss and there was little difference in Mg/Ca ratios we used this technique for our study.





170 In experiment 2, as a result of the low Ca concentrations in experiment 1 (0.55 to 7.50 ppm, Table 1A and A2 in Appendix A), the chemical cleaning procedure was amended to assess if this was due to calcite dissolution (too rigorous cleaning). In experiment 2, from 8 sets of 20 specimens of *Globigerinoides ruber*, picked from core NIOP929 (Table A1 in Appendix A), four sets were manually crushed using a drop of MilliQ water and a metal pin in a glass mortar. Two of those samples were transferred to a small glass vial and ultrasonicated for 3 seconds. The other two samples were kept intact and transferred to

175 Eppindorf tubes. The cleaning procedure followed Barker et al. (2003) except for reducing the time during methanol washes (20 seconds repeated twice compared to 1-2 min repeated once).

Generally, in experiment 2, Ca concentrations are higher than in experiment 1 (range from 7.32 to 30.92 ppm, see Table A2 in Appendix A and Fig. 4). In crushed samples, average Ca concentrations of 15.96 ppm (range from 7.92 to 22.54 ppm) are lower than the average in non-crushed samples of 23.16 ppm (range from 7.32 to 30.92 ppm), suggesting more sample loss

180 from crushing. Because samples that were crushed have a lower average Fe/Ca ratio than the uncrushed samples (0.38 mmol mol$^{-1}$ compared to 0.61 mmol mol$^{-1}$) this suggests less silicate contamination in the crushed samples. There is, however, significant uncertainty because the offset is not consistent and one outlier with a substantially higher Fe/Ca ratio (1.56 mmol mol$^{-1}$) and Mg/Ca ratio (9.41 mmol mol$^{-1}$) is partly responsible for the higher average (see Figure 4). It is interesting to note that there is only a small difference in average Al/Ca ratios (2.91 and 3.01 mmol mol$^{-1}$) with no correlation with Mg/Ca ratios

185 (Figure 4b) which suggest no difference in silicate contamination. The strong correlation between Mn/Ca ratios and Mg/Ca ratios in both crushed and uncrushed tests suggest insufficient removal of Mn-oxide coatings in both. This finding supports the notion that, regarding Mn-oxide coatings, the rigour of the chemical cleaning is significantly more important than mechanical crushing. Overall, the outcomes from experiment 2 suggest that crushing of foraminifera ensures better cleaning results. Due to time constraints we decided to crush shells prior to chemical cleaning in the subsequent experiment 3 following previous

190 benthic core-top studies (Elmore et al., 2015; Elderfield et al., 2010; Barker et al., 2003).

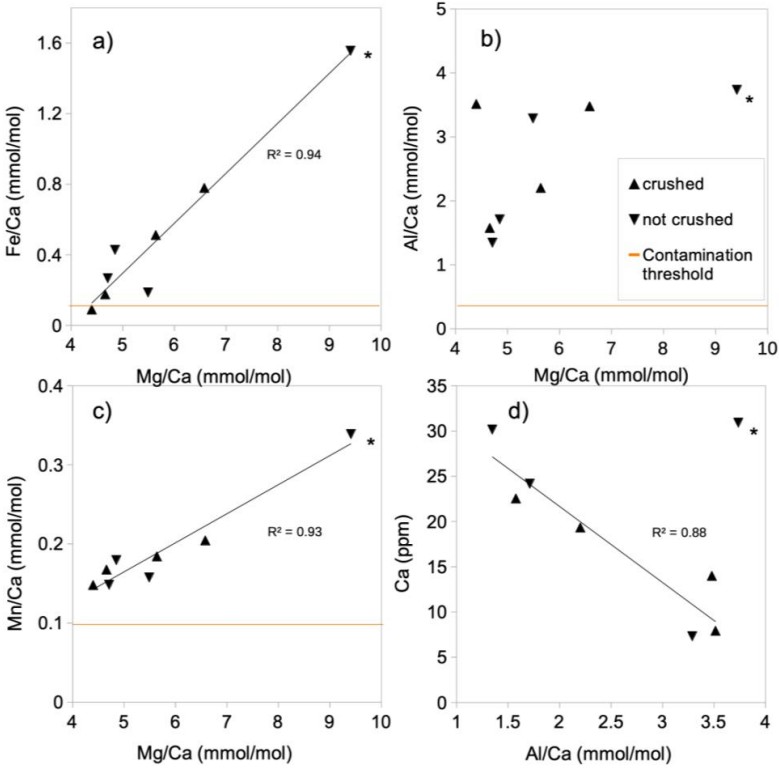



**Fig. 4** Mg/Ca ratios in *G. ruber* in NIOP929 (Saher et al., 2009) using procedure of Experiment 2. **a.** Correlation between Fe/Ca ratios and Mg/Ca ratios. See comments in text regarding sample with high Fe/Ca and Mg/Ca ratios. With this sample being include, $R^2$ is 0.94. Without this sample $R^2$ is 0.72. **b.** no correlation between Al/Ca ratios and Mg/Ca ratios. **c.** Correlation between Mn/Ca ratios and Mg/Ca ratios ($R^2$=0.93 with outlier and $R^2$=0.65 without **d.** Correlation between Ca concentrations and Al/Ca ratios, $R^2$=0.88 when outlier * is excluded.

In Experiment 3 the cleaning procedure followed Barker et al. (2003) with an additional 20 seconds total time of methanol wash added based on the contamination post-cleaning identified in Experiment 2 (35 seconds repeated thrice which closely follow Barker et al.'s (2003) total time of 1-2 min repeated twice). In the procedure, specimens of *Globigerinoides ruber* (6x25) picked from core NIOP929, *Uvigerina. spp.* (10) from core PE303-17a, and *Cibicidoides spp.* (2x5) from cores PE303-17Aa and PE303-13b (Appendix A Table A1) were used.

In contrast to Experiment 2, there is no correlation between Mg/Ca ratios and Al/Ca (Figure 5a) and only a weak correlation between Mg/Ca ratios and Mn/Ca (Figures 5c) and most samples have Fe concentrations below limit of detection (<0.0058 ppm). This suggests no or minimal silicate contamination and Mn-oxide coatings (Barker et al., 2003; Elderfield et al., 2010). The average Al/Ca ratios in samples containing Al concentrations above the limit of detection (6 out of 9) is significantly above the threshold for contamination (>0.4 mmol mol$^{-1}$), 2.27 mmol mol$^{-1}$ (Figure 5a, Table A3 in Appendix A) but because there is no correlation with Mg/Ca ratios this could be due to issues with precision of measurements as reported by Elderfield et al., (2010) or due to contamination from contaminants other than silicate. The Al/Ca ratios showed a negative exponential correlation with Ca concentrations (Figure 5b) where high Al/Ca ratios correlated with low Ca concentration. This could suggest that there is a threshold for minimum Ca concentration for contaminants to accurately be determined. According to the exponential relationship of Al/Ca ratios and Ca concentrations in Fig. 5b, it is suggested that there is a threshold for total Ca concentration between 15 and 25 ppm. The average Mn/Ca ratio was 0.16 mmol mol$^{-1}$ (Figure 5c) which is slightly above the contamination threshold >0.1 mmol mol$^{-1}$, but below the contamination threshold of 0.2 mmol mol$^{-1}$ proposed by Hasenfratz et al. (2017) suggesting that Mn-oxide coatings have an insignificant effect on Mg/Ca ratios.

Based on the results from Experiment 3, indicating no or minor silicate contamination and Mn-oxide coatings, the cleaning methodology of core-tops followed the methodology used in Experiment 3, i.e. Barker et al. (2003) .





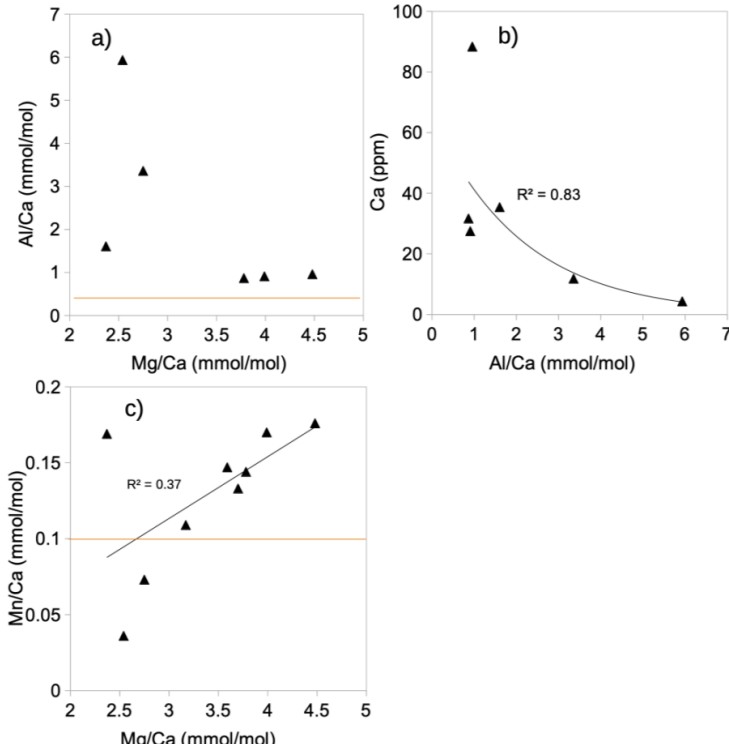

**Fig. 5.** Correlation between **a.** Al/Ca ratios and Mg/Ca ratios, **b.** Ca concentration and Al/Ca ratios, **c.** Mn/Ca ratios and Mg/Ca ratios of samples of *G. ruber* from core NIOP929 following procedure in Experiment 3. Orange horizontal line: Al/Ca and Mn/Ca contamination thresholds (0.4 and 0.1 mmol/mol respectively).

### 2.3.    Mg/Ca analysis in ICP-OES

Supernatants (300 µl) of the dissolved samples were transferred to acid-cleaned polypropylene tubes and diluted to 1.8 ml with dilute $HNO_3$. Samples were analysed in an ICP-OES Varian Vista Pro at the Grant Institute at The University of Edinburgh.

Emission intensity was normalised to concentrations using a standard calibration curve based on measurements of high purity standards (element calibrations and concentration of standards in Figure 1 and Table B1 in Appendix B). Analytical lines (Mg) 285 and (Ca) 315 are used as these are reported to minimise matrix effect (De Vielliers et al., 2002). Instrumental precision was ± 1% based on 6 replicate measurements of ECRM 752-1 carbonate standard. The limit of detection was calculated by 3 multiples of the standard deviation. The calibration curves of the standards have an $r^2 > 0.99$ (Appendix B Figure B1). To overcome matrix effect associated with Ca analysis, a calibration was produced using standard solutions of increasing Mg/Ca ratios (Appendix B Table B1) as described by De Vielliers et al. (2002). In addition, the ECRM was diluted to a concentration of 40 ppm Ca (similar to the concentrations of the samples studied) which also assumes a similar matrix effect. Two procedural blanks and two samples of ECRM-752-1 carbonate standard with Mg/Ca ratio 3.762 mmol mol[-1] were included in every run. In addition to Mg/Ca, Fe/Ca, Al/Ca and Mn/Ca were calculated to monitor silicate contamination and Mn-oxide coatings (Barker et al., 2003).

Following Barker et al. (2003) and Elderfield et al. (2010) contamination thresholds of Fe/Ca, Al/Ca and Mn/Ca ratios used were 0.1, 0.4 and 0.1 mmol mol[-1], respectively. Linear regression was plotted between Fe/Ca, Al/Ca, Mn/Ca ratios and Mg/Ca



ratios and the r² value was used to assess if there was a significant correlation. Procedural blanks were used to correct Mg, Ca, Fe, Al and Mn concentrations for any introduced contaminants. Mg/Ca ratios were calculated using Ca315 and Mg285 concentrations in ppm and ppb, corrected by blanks and converted to mmol mol⁻¹ by:

$$\text{Mg/Ca} = \frac{(Mg285\,(ppb) - Mg285_{blank})/M_{Mg}}{(Ca315\,(ppm) - Ca315_{blank})/M_{Ca}}$$

where $M_{Mg}$ and $M_{Ca}$ refers to the atomic masses of Mg (24.305 g mol⁻¹) and Ca (40.08 g mol⁻¹). This was also used for Fe/Ca, Al/Ca, Mn/Ca using their respective atomic masses ($M_{Fe}$, $M_{Al}$ and $M_{Mn}$).

### 2.4.    Mg/Ca – BWT calibration

A linear regression was applied to assess correlation between the Mg/Ca ratios measured in *Cibicidoides wuellerstorfi*,
*Cibicidoides mundulus*, *Uvigerina peregrina* and *Cibicidoides spp.* and modern bottom water temperature from the nearest hydrographic temperature profile (see Table 1). The slope of Mg/Ca ratios over BWT were compared with previous studies from the Atlantic, Pacific and Indian Ocean.

### 3.    RESULTS

#### 3.1.    Core-top Mg/Ca ratios in *Cibicidoides spp.*

In samples (Table 1) containing *Cibicidoides wuellerstorfi* and *Cibicidoides mundulus* the Ca concentrations range from 4.00 to 68.72 ppm (see Table 2). Mg/Ca ratios range from 1.19 to 6.04 ± 0.03 mmol mol⁻¹ in *Cibicidoides spp.* (12 samples), and 3.17 to 4.18 ± 0.05 mmol mol⁻¹ in *G. ruber* (6 samples). The Fe/Ca ratios range from 0.13 to 0.35 mmol mol⁻¹ in *Cibicidoides wuellerstorfi*, from 0.98 to 1.10 mmol mol⁻¹ in *Cibicidoides mundulus* and 0.08 to 2.45 mmol mol⁻¹ in *Cibicidoides spp*. All six samples of *G. ruber* (analysed in the same run, from core NIOP929) have Fe concentrations below the limit of detection
(<0.0034 ppm, see Table A3 in Appendix A). The Al/Ca ratios range from 0.28 to 0.57 mmol mol⁻¹ in *Cibicidoides wuellerstorfi*, from 0.24 to 2.66 mmol mol⁻¹ in *Cibicidoides mundulus,* from 0.21 to 0.36 mmol mol⁻¹ in *Cibicidoides spp*. (Table 2) and 0.25 and 0.37 mmol mol⁻¹ in *G. ruber* (Table A3 in Appendix A). The Mn/Ca ratios range from 0.01 to 0.20 mmol mol⁻¹ in *Cibicidoides spp.* samples (Table 2) and from 0.13 to 0.19 mmol mol⁻¹ in the planktic samples (Table A3 in Appendix A). There is no obvious correlation between the Fe/Ca, Al/Ca and Mn/Ca with Mg/Ca ratios for any of the
Cibicidoides species, except for *Cibicidoides mundulus* (Figure 6a, b, c). In this figure, although mostly driven by one possible outlier, a correlation of Al/Ca with Mg/Ca might be indicated. Regarding contamination thresholds of 0.1 , 0.4 and 0.1 mmol mol⁻¹ for  the Fe/Ca, Al/Ca and Mn/Ca ratios, respectively, all *Cibicidoides spp.* samples are below the threshold for Al/Ca and Mn/Ca ratios. In relation to the Fe/Ca ratio, one sample was below, one sample just above and three samples well above the contamination threshold (Table 2).





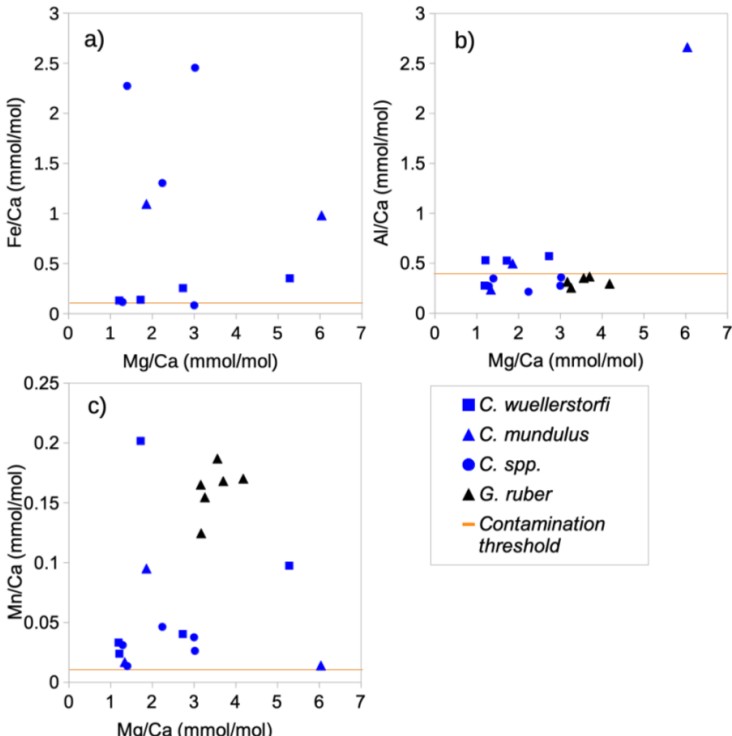


**Fig. 6.** Correlation between **a.** Fe/Ca ratios, **b.** Al/Ca ratios, **c.** Mn/Ca ratios and Mg/Ca ratios in *Cibicidoides spp.* including *C. wuellerstorfi* and *C. mundulus* and *G. ruber* (control group). **d.** The correlation between total Ca concentration and contamination (Fe/Ca ratios). The horizontal line show threshold at 0.1, 0.4 and 0.1 mmol mol$^{-1}$ as an indication of contamination.

  





**Tab. 2** Mg/Ca ratios, contamination indicators (Fe/Ca, Al/Ca and Mn/Ca ratios) and Ca measured in *C. wuellerstorfi*, *C. spp.*
*C. mundulus* and *U. peregrina* from Tanzania core-top samples.

| Core | Species | Mg/Ca | Fe/Ca | Al/Ca | Mn/Ca | Ca | Note |
|------|---------|-------|-------|-------|-------|-----|------|
| PE303-4c | *C. wuellerstorfi* | 1.19 | <LOD | 0.28 | 0.03 | 16.05 | |
| PE303-6a | *C. wuellerstorfi* | 1.72 | 0.14* | 0.53* | 0.20* | 35.09 | |
| PE303-3 | *C. wuellerstorfi* | 1.21 | 0.13* | 0.53* | 0.02 | 68.73 | |
| PE303-22 | *C. wuellerstorfi* | 5.28 | 0.35* | <LOD | 0.10 | 3.99 | ? |
| PE303-18b | *C. wuellerstorfi* | 2.73 | 0.25* | 0.57* | 0.04 | 14.43 | ? |
| PE303-6a | *C. mundulus* | 1.86 | 1.09* | 0.49* | 0.10 | 20.56 | e |
| PE303-3 | *C. mundulus* | 1.34 | <LOD | 0.23 | 0.02 | 15.84 | |
| PE303-18b | *C. mundulus* | 6.04 | 0.98* | 2.66* | 0.01 | 55.91 | e |
| PE303-3 | *C. spp.* | 1.40 | 2.27* | 0.34 | 0.01 | 17.27 | e |
| PE303-14a | *C. spp.* | 1.29 | 0.12* | 0.27 | 0.03 | 75.62 | |
| PE304-9 | *C. spp.* | 3.02 | 2.45* | 0.35 | 0.03 | 30.96 | e |
| PE304-30c | *C. spp.* | 2.24 | 1.30* | 0.21 | 0.05 | 22.22 | e |
| PE303-50 | *C. spp.* | 3.00 | 0.08 | 0.27 | 0.04 | 28.53 | |
| PE303-6a | *U. peregrina* | 1.17 | 0.02 | 0.91* | 0.05 | 18.54 | |
| PE303-14a | *U. peregrina* | 1.10 | 0.15* | 0.68* | 0.009 | 17.61 | e |
| PE303-6a | *U. peregrina* | 2.17 | 0.67* | 1.09* | 0.07 | 16.28 | e |
| PE303-4c | *U. peregrina* | 1.58 | 0.07 | 0.71* | 0.03 | 20.35 | |
| PE303-17a | *U. peregrina* | 2.99 | 1.66* | 2.34* | 0.06 | 41.28 | e |
| PE304-9 | *U. peregrina* | 1.82 | 0.43* | 1.42* | 0.009 | 68.17 | e |
| PE303-18b (1) | *U. peregrina* | 2.76 | 2.02* | 4.02* | 0.02 | 79.35 | e |
| PE303-18b (2) | *U. peregrina* | 2.52 | 1.55* | 3.90* | 0.02 | 74.32 | e |
| PE304-25 | *U. peregrina* | 2.69 | 2.04* | 3.61* | 0.06 | 65.80 | e |

*above contamination threshold 0.1, 0.4 and 0.1 for Fe/Ca, Al/Ca and Mn/Ca (Elderfield et al., 2010)

**clear outlier based on typical Mg/Ca range reported in previous studies

e = excluded due to high contamination, ? = ambiguous assessment of contamination

<LOD = below limit of detection

### 3.2. Core-top Mg/Ca ratios in *Uvigerina peregrina*

In the core-top samples (Table 1) the Ca concentration in *Uvigerina peregrina*, range from 16.28 to 74.32 ppm (Table 2). The
Mg/Ca ratios vary between 1.10 and 2.99 mmol mol$^{-1}$ ± 0.02 mmol mol$^{-1}$ in *Uvigerina peregrina* (9 samples), and 4.77 and
5.22 ± 0.05 mmol mol$^{-1}$ in *G. ruber* (3 samples; Figure 7 and Table 2). The Fe/Ca ratios range from 0.02 to 2.04 mmol mol$^{-1}$ in



*Uvigerina peregrina* and 0.24 to 0.38 mmol mol[-1] in *G. ruber*. The Al/Ca ratios are between 0.68 and 4.02 mmol mol[-1] in *Uvigerina peregrina* and 0.92 to 1.31 mmol mol[-1] in *G. ruber*. The Mn/Ca ratios vary between 0.01 and 0.08 mmol mol[-1] in *Uvigerina peregrina* and 0.11 and 0.14 mol mol[-1] in *G. ruber*. There is a strong positive correlation ($r^2$ = 0.87) between the Fe/Ca ratios and the Mg/Ca ratios (Figure 7a.) and a positive correlation ($r^2$ = 0.66) between the Al/Ca ratios and the Mg/Ca ratios (Figure 7b.). There is no correlation between the Mn/Ca ratios and the Mg/Ca ratios (Figure 7c). Regarding contamination, all *Uvigerina peregrina* samples are below the threshold for Mn/Ca ratios. All samples are above the threshold for Al/Ca ratios, some rather narrowly so. Two samples were below the threshold for Fe/Ca, one narrowly above and the remaining 6 samples partially well above the limit (Table 2).

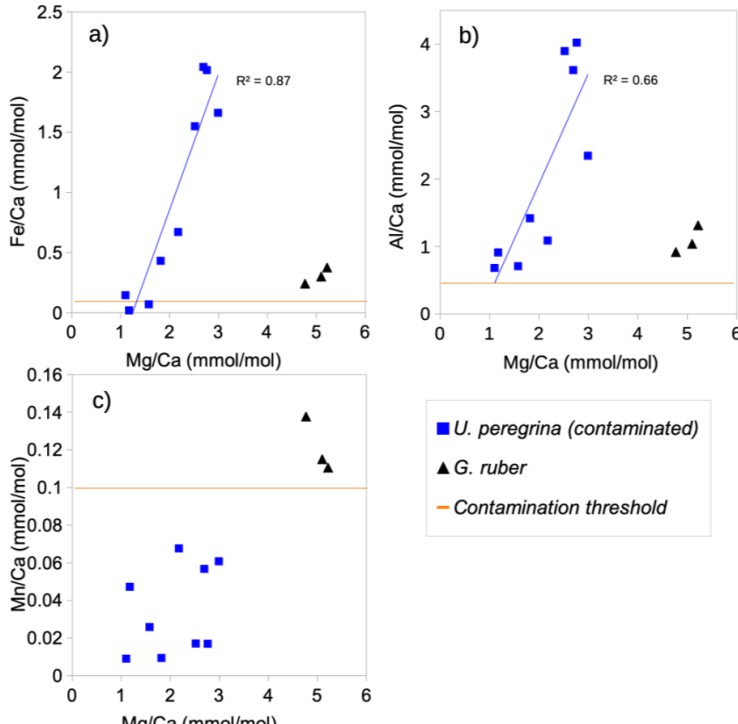

**Fig. 7.** The correlation between the Mg/Ca ratios and **a.** Fe/Ca ratios, **b.** Al/Ca ratios and **c.** Mn/Ca ratios in blue: *Uvigerina peregrina* and black: *G. ruber* (control group). Horizontal line show Fe/Ca, Al/Ca and Mn/Ca contamination thresholds (0.1, 0.4 and 0.1 mmol mol[-1]). Values below the limit of detection are not plotted. Trendline in a. and b. with $R^2$ show linear correlation of *Uvigerina peregrina*.

### 3.3. Correlation between contamination and core-top depth

Figure 8a-b shows Fe/Ca and Al/Ca ratios from *Uvigerina peregrina* and *Cibicidoides spp.* versus the retrieval depth of the samples. There is no correlation between Fe/Ca- or the Al/Ca ratios and depth for *Cibicidoides. spp* while *Uvigerina peregrina* displays such correlations with $r^2$ values of 0.58 and 0.65, respectively. Below 2500 m, *Uvigerina peregrina* (five samples) shows significantly smaller average Fe/Ca and Al/Ca ratios (0.23 and 0.85 mmol mol[-1]) compared samples from <1500 m (four samples) (1.54 and 3.06 mmol mol[-1]).



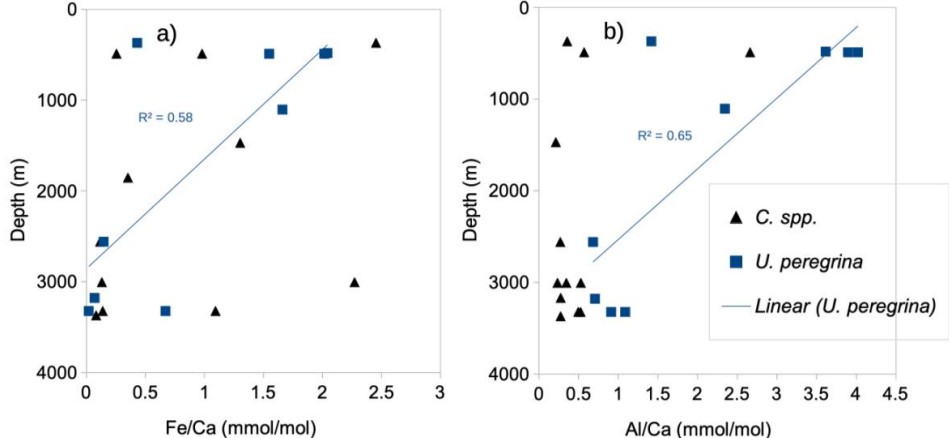

**Fig. 8.** Correlation between water depth of sediment surface samples and contamination (**a.** Fe/Ca and **b.** Al/Ca) in *Uvigerina peregrina* and Cibicidoides species.

### 3.4.   Mg/Ca – BWT calibration

Principally we used the thresholds of 0.1, 0.4 and 0.1 mmol mol$^{-1}$ of Fe/Ca, Al/Ca and Mn/Ca ratios following Elderfield et al. (2010) and Barker et al. (2003) as well as correlations between Fe/Ca, Al/Ca and Mn/Ca ratios with Mg/Ca ratios following Barker et al. (2003) to assess silicate and/or Mn-oxide contamination. All but two samples of *Uvigerina peregrina* (Table 2) were excluded due to high Fe/Ca ratios (>0.1 mmol mol$^{-1}$) and a strong correlation with Mg/Ca (Figure 7a). Mg/Ca ratios of *Cibicidoides spp.* samples with Fe/Ca ratios >0.1 and <0.3 mmol mol$^{-1}$ were included which showed no correlation between

Fe/Ca ratios and Mg/Ca ratios (Table  2, figure 6). The Mg/Ca ratios not included in core-top calibration are in Table 2 (annotated 'e').

The Mg/Ca ratios of *Cibicidoides spp.*, *C. mundulus* and *Cibicidoides wuellerstorfi* (Table 2) were plotted versus BWT (temperature profiles from positions close to our core-top transect from Birch et al., 2013; figure 9). For *Cibicidoides. spp*, *Uvigerina peregrina* and *Cibicidoides mundulus* discerning robust relationships between the Mg/Ca relationships and BTW is

not straightforward. Based on the no-correlation argument above, and ignoring contamination thresholds, tentative relationships are indicated for *Cibicidoides. spp* and *Cibicidoides mundulus*. These are, however, partially based on samples with signs of contamination being reflected in the Fe/Ca and/or the Al/Ca ratios. Removing those sample entails an insufficient amount of data remaining to establish a relationship (see figure 9). For *Cibicidoides wuellerstorfi* there is little indication of strong contamination. Only some Al/Ca ratios are slightly above the contamination threshold. Establishing a straightforward

relationship of Mg/Ca with BTW is hampered by one sample with unusually high Mg/Ca values. We regard this sample as an outlier for unknown reason. Figure 9 shows the resulting relationship for *Cibicidoides wuellerstorfi (see formula below)* alongside the remaining samples for the other species. In Fig. 9 the linear correlation for *Cibicidoides wuellerstorfi*  is:

$$Mg/Ca = 0.19 \pm 0.02 * BWT + 1.07 \pm 0.03, r^2 = 0.87$$





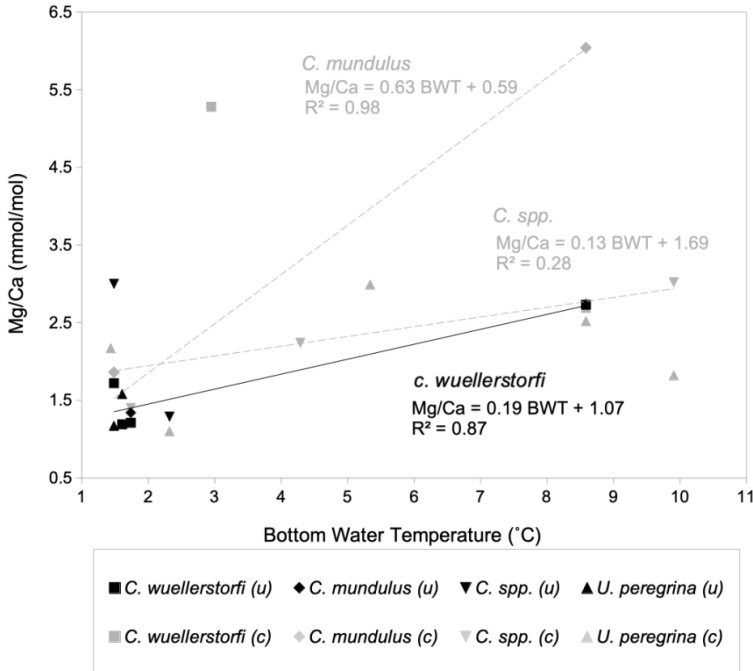

**Fig. 9.** Mg/Ca ratios of *Cibicidoides spp.*, *Cibicidoides wuellerstorfi*, *Cibicidoides mundulus* and *Uvigerina peregrina* over bottom water temperature (BWT). Grey: Measurements with suspected significant contamination (c). Black: uncontaminated/minor contamination based on contamination thresholds (0.1 mmol/mol Fe/Ca, 0.4 mmol/mol Al/Ca and 0.1 mmol/mol Mn/Ca) and correlations with Mg/Ca. There are several data points where presence of contamination is ambiguous - see discussion. Black trendline represent Mg/Ca-BWT linear correlation in *C. wuellerstorfi* (four samples with minor contamination), grey trendlines represent Mg/Ca-BWT linear correlations in *C.mundulus* and *C.spp.* respectively (contaminated and uncontaminated samples included).

## 4. DISCUSSION

### 4.1. Mg/Ca values in *Cibicidoides:* data quality and core top calibrations

All samples of *Cibicidoides* except for one have Mn/Ca ratios below the threshold for contamination (<0.1 mmol mol[-1]) and no correlation with Mg/Ca ratios, suggesting no or insignificant Mn-oxide coatings (Hasenfratz et al., 2017). Based on silicate contamination indicated by Fe/Ca and Al/Ca ratios being significantly above the contamination threshold, six samples were excluded (three *Cibicidoides. spp*, two *Cibicidoides mundulus* and one *Cibicidoides wuellerstorfi* samples) (Table 2). When plotted at the genus level, *Cibicidoides* data show no correlation between Fe/Ca or Al/Ca ratios and Mg/Ca ratios (Figure 6a-b) supporting the notion of no silicate contamination, amid this strategy being in line with previous approaches (e.g. Elderfield et al. 2006, Healey et al., 2008). When plotted at a species level, however, there is a strong correlation ($r^2$=0.94) between Fe/Ca and Mg/Ca ratios for *Cibicidoides wuellerstorfi* data (Figure A1 in Appendix A) which suggests silicate contamination. The indicated contamination levels are small in most cases with Fe/Ca ratios below 0.25 mmol mol[-1]. It is difficult to assess how much this small contamination has affected the Mg/Ca data. If the increase in Mg/Ca ratios from silicate contamination is



within the uncertainty of Mg/Ca ratio determinations (~0.03 mmol mol$^{-1}$) this can be neglected. We therefore used the Mg/Ca ratios of four *Cibicidoides wuellerstorfi* samples with Fe/Ca below 0.25 mmol mol$^{-1}$ to establish a Mg/Ca BWT relationship (Table 2, Figure 9).

There are two abnormally high Mg/Ca ratios measured in *Cibicidoides mundulus* (6.04 mmol mol$^{-1}$) and *Cibicidoides wuellerstorfi* (5.28 mmol mol$^{-1}$; Table 2) compared to Mg/Ca ratios in some studies (Elderfield et al., 2006; Rosenthal 1997)

but within range of Mg/Ca ratios in other reports (Lear 2002; Rosenthal 1997). The *C. mundulus* sample with a Mg/Ca ratio of 6.04 mmol mol$^{-1}$ shows a broadly similar Fe/Ca ratio but a significantly higher Al/Ca ratio (2.66 mmol mol$^{-1}$) than other measurements from the genus *Cibicidoides* (Al/Ca ratios ranging from 0.21 to 0.57 mmol mol$^{-1}$; Table 2). It is uncertain whether the high Mg/Ca ratio is a result from silicate contamination or is due to another Mg bearing contaminant also high in Al. The Mn/Ca ratio in this sample is low (0.01 mmol mol$^{-1}$) indicating no presence of Mn-oxide coatings.

The *C. wuellerstorfi* sample with a Mg/Ca ratio of 5.28 mmol mol$^{-1}$ does not have significantly higher Al/Ca, Fe/Ca or Mn/Ca ratios compared to other samples of *Cibicidoides spp.* but has significantly less Ca (3.99 ppm compared to 14.42 to 75.62 ppm). The results from Experiment 3 implied a minimum concentration for Ca around 15 ppm for reliable Mg/Ca measurements. In this sample, the Ca concentration is significantly lower than the threshold which could be responsible for the abnormally high Mg/Ca ratio. The low Ca content could be due to sample loss in crushing, sample loss during transfer in chemical cleaning or

calcite dissolution from chemical cleaning.

The Mg/Ca ratio-BWT relationship of *Cibicidoides wuellerstorfi* $Mg/Ca = 0.19 \pm 0.02 * BWT + 1.07 \pm 0.03$, indicate increasing Mg/Ca ratios with increasing temperature, and is broadly consistent with previous studies (Figure 10, Healey et al., 2008; Lear et al., 2002) although there is only one data point reflecting the high temperature end of our calibration range of 3-8 ˚C. The temperature sensitivity of *Cibicidoides wuellerstorfi* in this study is lower than in Elderfield et al. (2006) and Healey

et al. (2008); 19% increase per 1˚C change in temperature compared to 30% (Healey et al., 2008) and 46 and 52% (Elderfield et al., 2006), see Fig. 10. Also, two Mg/Ca ratios (ignoring a probable outlier with a high Mg/Ca ratio) at lower temperatures (<2˚C) are within the data range of both, the southeast Indian Ocean calibration by Healey et al. (2008) and the Southwest Indian Ocean calibration by Elderfield et al. (2006) but are higher than the Mg/Ca ratios from the Somali basin (included in Elderfield 2006 data in Figure ???11). These high Mg/Ca values do, however, fall within the range of values found in

*Cibicidoides wuellerstorfi* from the Atlantic Ocean. There is a discussion surrounding relatively unmixed NADW crossing the Davie Ridge into the Somali basin (van Aken et al., 2004). Our core-top sample set is in the flow path of NADW, supporting the notion our high Mg/Ca ratios reflecting NADW specific water properties. Firm conclusions are hampered by the limited sample size in our *Cibicidoides wuellerstorfi* data set. If correct, however, changing water masses in a given location, may add additional uncertainties to BWT reconstructions.



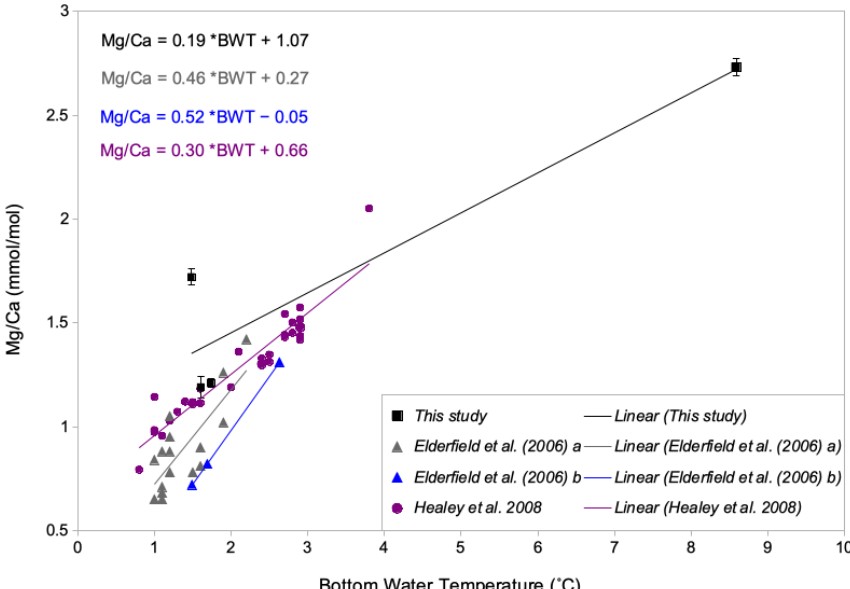

**Fig. 10.** Mg/Ca – BWT calibration of *Cibicidoides wuellerstorfi* in black: this study with error bars showing standard deviation, purple: S. E Indian Ocean from Healey et al. (2008), grey: S. W Indian Ocean and blue: Somali basin from Elderfield et al. (2006).

Previous studies have used both linear and exponential regressions to describe the temperature dependence of Mg/Ca ratios (e.g. Healey et al., 2008; Lear et al., 2002, Martin and Lea, 2002; Elderfield et al., 2006) with some studies suggesting the latter being preferable at low temperatures and over narrow temperature ranges (Healey et al., 2008; Stirpe et al., 2021). The small sample size in our study hampers assessment of the better regression strategy. The generally good fit with the linear regression in Healey et al. (2008) and the data ranges in Lear et al. (2002), support the notion of our Indian Mg/Ca calibration being broadly correct.

Two out of three Fe/Ca and Al/Ca ratios for *Cibicidoides mundulus* are significantly above contamination thresholds (Table 2). In the absence of a correlation with Mg/Ca ratios (Figure 6a-b) all three Mg/Ca ratios were tentatively plotted and compared to existing *Cibicidoides mundulus* core-top calibrations (Figure 11). The estimated Mg/Ca ratios in the temperature range of 1-2˚C is within the range of Healey et al. (2008). One of the data points seems sufficiently cleaned whilst the second does not, based on low and high Al/Ca and Fe/Ca ratios, respectively. Because both values lie within the range of data provided by Healey et al. (2008) this suggests high estimates of Fe/Ca and Al/Ca ratios being a result of a non-Mg bearing contaminant (not silicate), supported by absent correlations between Al/Ca or Fe/Ca with Mg/Ca (Figure 6a-b). Alternatively, this could suggest increased Mg/Ca ratios that maybe interpreted as silicate contamination but are within the natural variation of Mg/Ca ratios in *Cibicidoides mundulus*.



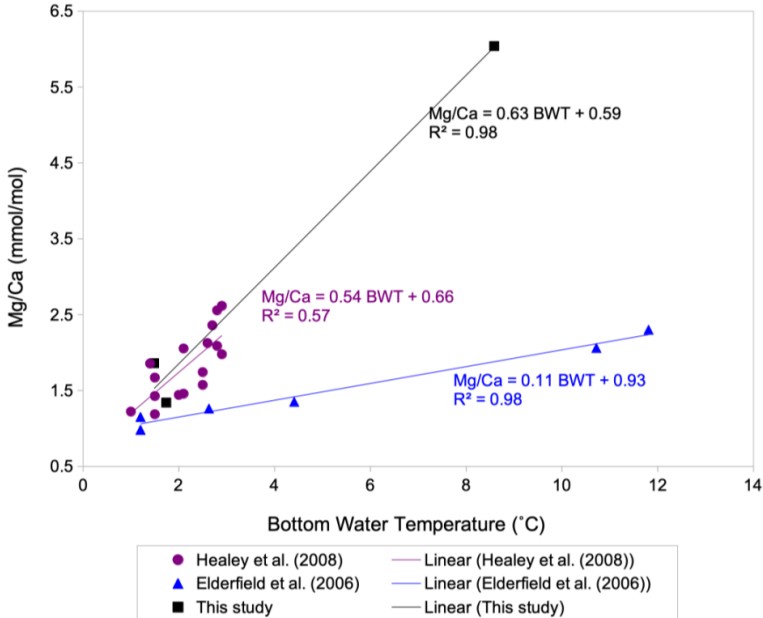

**Fig. 11.** Mg/Ca – BWT calibration of *Cibicidoides mundulus* in this study (black) compared to core-top calibrations in *Cibicidoides mundulus* by Healey et al. (2008) (purple) and Elderfield et al. (2006) (blue).

The linear relationship of the three Mg/Ca ratios of *Cibicidoides mundulus* in Fig. 11 closely resembles the linear relationship derived from data by Healey et al. (2008) from core-top estimates from the Atlantic, Pacific and Indian Ocean combined (Figure 11) although the reliability in our study is limited by only three datapoints and only one value at high temperatures. Our and Healey et al's (2008) calibrations differ from the S.W Indian Ocean calibration from Elderfield et al. (2006) with the reasons for this discrepancy being unclear.

### 4.2.   Mg/Ca ratios in Uvigerina peregrina

Our results for *Uvigerina peregrina* show that Mg/Ca ratios in nine samples of *Uvigerina peregrina* range from 1.10 to 2.99 ± 0.02 mmol mol$^{-1}$ (Table 2) covering a depth range of 370-3323 m (Table 1). These Mg/Ca ratios are higher than values reported by Stirpe et al. (2021) ranging from 0.68 to 1.50 mmol mol$^{-1}$ covering a depth range of 663 to 4375 m. In our data set, 7 out of 9 samples have Fe/Ca ratios above the contamination threshold (>0.1 mmol mol$^{-1}$) and correlate positively with Mg/Ca ratios (r$^2$=87, Figure 7a) suggesting silicate contamination. Al/Ca ratios in all samples are above contamination threshold (>0.4 mmol mol$^{-1}$) and correlate with Mg/Ca ratios (r$^2$=0.66, Figure 7b). These findings suggest silicate contamination being reflected in our high Mg/Ca ratios. Mn/Ca ratios in all samples are below contamination threshold (<0.1 mmol mol$^{-1}$) which suggest no presence of Mn-oxide coatings (Figure 7c).

To investigate if the high Mg/Ca ratios are indeed a result of silicate contamination these were plotted versus bottom water temperatures and compared to previous studies (Figure 12). Only two samples of *Uvigerina peregrina* are below the contamination threshold of Fe/Ca ratios (<0.1 mmol mol$^{-1}$, Elderfield et al., 2010). These map onto the relationship of *Uvigerina peregrina* by Elderfield et al. (2006). Most of the samples with Mg/Ca ratios that were suggested to be silicate contaminated are, as expected, significantly higher than previous estimates (Figure 12), up to 1.5 mmol mol$^{-1}$ higher than in the relationship of Elderfield et al. (2006). This supports the notion that Fe/Ca and Al/Ca ratios well above the contamination threshold indeed identify samples with contamination that bias the Mg/Ca ratios.



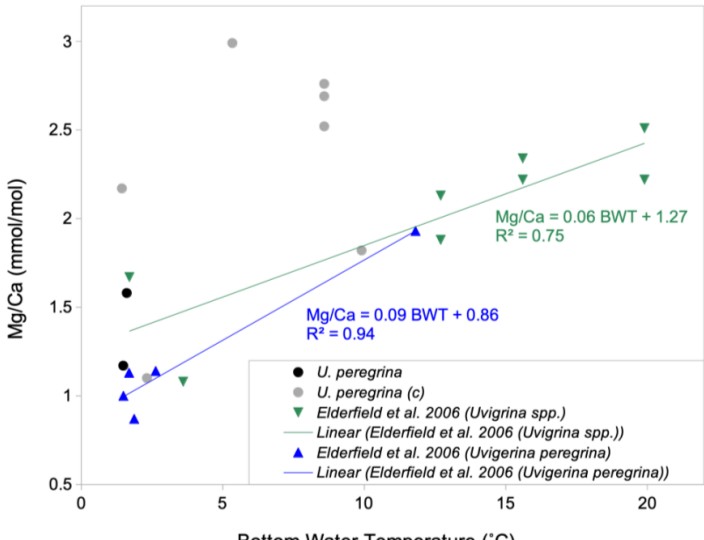

**Fig. 12** Mg/Ca – BWT calibration of *Uvigerina peregrina* in this study (circles, grey: contaminated, black: uncontaminated) compared to Elderfield et al. (2006) core-top calibrations of blue: *Uvigerina peregrina* and green: *Uvigerina spp*.

Mg/Ca ratios measured in *Uvigerina peregrina* in a previous study (Yu et al., 2007) showed no significant difference between cleaning method using weaker reductive cleaning reagents and oxidative cleaning only, in contrast to Mg/Ca ratios measured in *Cibicidoides wuellerstorfi* and *Cibicidoides mundulus* showing a significant difference (Yu et al., 2007). The clear difference in contamination, between *Cibicidoides spp*. (Figure 6 and *Uvigerina peregrina* (Figure 7) despite using the same cleaning procedure, supports the findings in (Yu et al., 2007), which suggest different rigour might be required for different species (please see section on variable degree of contamination).

### 4.3. Sufficient cleaning of Mn-oxide coatings

While the cleaning procedure used in this study by Barker et al. (2003) has been widely used (e.g. Elderfield et al., 2006; Elderfield et al., 2010, Elmore et al., 2015), inefficient removal of Mn-Mg coatings has been observed (Hasenfratz et al., 2017; Pena et al., 2005). The Mn-oxide coatings which are found on the inner shells of foraminifera can cause increased Mg/Ca ratios and only the reductive cleaning procedure satisfactorily removes this (Pena et al., 2005). Where Mn/Ca ratios are below 0.2 mmol mol$^{-1}$, it entails a small increase in Mg/Ca ratios that is within the uncertainty of Mg/Ca ratio determination and therefore can be considered insignificant (Hasenfratz et al., 2017). All but one core-top sample have Mn/Ca ratios below 0.2 mmol mol$^{-1}$ (Figure 6 and 7, Table 2). This suggests the reductive cleaning step was not needed for samples analysed in this study, and therefore it is assumed the 'Mg cleaning procedure' utilised in this study is more suitable than the 'Cd cleaning procedure'.

### 4.4. Inefficient cleaning of silicate contaminants

The Fe/Ca and Al/Ca ratios in all but two samples of the *Uvigerina peregrina* and six samples of *Cibicidoides spp*. (Table 2) suggest inefficient removal of silicate contaminants, implying that the number of rinse/ultrasonication repeats was insufficient for efficient cleaning of samples despite following the Barker et al. (2003) procedure. Increasing the number of rinse/ultrasonication repeats further (from four to five) entails the risk of considerable calcite dissolution which may lower the



Mg/Ca ratios (Marr et al. 2013). There is probably a threshold at which tests are thoroughly cleaned and tests dissolve. A stepwise leaching test series could be used to investigate the rigour needed to optimise cleaning whilst avoiding sample loss in

the process. Due to time limitations, however, this was not possible. If the methodology needs to be adapted to specific foraminifera species this highly limits the comparability between studies investigating different species from different core locations.

### 4.5. Species specific differences in silicate contamination

The range of Fe/Ca ratios in *Uvigerina peregrina* was higher (0.43 to 2.04 mmol mol$^{-1}$) than in *Cibicidoides wuellerstorfi* (0.13

to 0.35 mmol mol$^{-1}$). This is consistent with Elmore et al. (supplementary material, 2015) reporting Fe/Ca ratios below 0.1 mmol mol$^{-1}$ in *Uvigerina peregrina* compared to below 0.04 mmol mol$^{-1}$ in *Cibicidoides wuellerstorfi*. Both ranges are below the contamination threshold (0.1 mmol mol$^{-1}$) in contrast to ranges reported in this study. Elmore et al. (2015) also used the procedure of Barker et al. (2003). Samples containing *G. ruber* from the core NIOP 905 were included in the analysis of core-tops and used as a control to monitor cleaning efficiency. On average the *G. ruber* contained Fe/Ca, Al/Ca and Mn/Ca ratios

of 0.18, 0.60 and 0.15 mmol mol$^{-1}$ (see Table A2 in Appendix A). Fe/Ca and Al/Ca ratios in samples of *G. ruber* from NIOP929 that were analysed in run along with *Uvigerina peregrina* (0.31 and 1.09 mmol mol$^{-1}$) were higher than the Fe/Ca and Al/Ca ratios in samples of *G. ruber* that were analysed in run alongside *Cibicidoides spp* (Table A2 in Appendix A). Since the same procedure was followed, the difference could point to an issue in the repeatability of the cleaning procedure, i.e. build-up of gas bubbles in hot water bath during the oxidative step, insufficient crushing prior to cleaning or different quantity of MilliQ

water and methanol removed in between rinses affecting efficiency of contaminants being removed in every rinse. Alternatively, the different contamination levels might be due different samples having been used as a result of insufficient specimens of *G. ruber* being contained in a single sample from core NIOP929.

### 4.6. Variable initial degree of contamination

The degree of contamination of tests depends on factors including sediment composition, sedimentation rates, oxygen, depth

in core, depth, and morphology (Barker et al., 2003; Ni et al., 2020; Pena et al., 2005). While foraminiferal tests that are well preserved and show no to minor signs of contamination were selected for analysis, the condition of core-top samples vary (qualitative observations of samples described in Appendix C). If *Uvigerina peregrina,* as an endobenthic species is subject to more contact with surrounding sediment particles than *Cibicidoides wuellerstorfi* this could explain higher contamination in *Uvigerina peregrina*. Also, the structure of the test wall may lead to different susceptibilities for contamination. By comparison,

the surface of tests of *Cibicidoides wuellerstorfi* are relatively smooth. The surfaces of *Uvigerina peregrina* tests, however, are irregular, entailing a larger surface area compared to *Cibicidoides wuellerstorfi* which in turn increases the probability of contaminants sticking onto tests of *Uvigerina peregrina*.

### 4.7. Different depositional environment

This study benefits from using core-top samples in a nearby region in contrast to previous studies (e.g. Elderfield et al., 2006).

While core-tops are located within a nearby region they cover a depth range of 370 m to 3323 m and are thereby in different depositional environments. Barker et al. (2003) suggest samples from regions of higher clay content require more rigorous cleaning. To investigate correlation between depositional environment and silicate contamination Fe/Ca, Al/Ca ratios over depth were plotted (see Figure 8. There is an inverse correlation between Fe/Ca and Al/Ca ratios with depth in *Uvigerina peregrina* samples (Figure 7, and samples at depths >2000 m have significantly lower Fe/Ca and Al/Ca ratios. Our core top

transect is located close to the Rovuma River, implying lithogenic material deposited near its mouth. The redistribution of these



sediments may well have affected the upper parts more than the deeper parts of the continental slope in our study area, which is probably reflected in the higher contamination level at shallower depths.

## 4.8. Relative impact of contamination

The Mg/Ca ratios are typically lower in benthic foraminifera compared to planktic foraminifera and therefore the relative impact of contamination in benthic foraminifera is larger (de Vielliers et al., 2002). While contamination thresholds following previous benthic foraminifera core-top studies have been used here, a lower contamination threshold for benthic foraminifera should be used to minimise the relatively higher uncertainty for benthic Mg/Ca ratios (Hasenfratz et al., 2017). Different species of benthic foraminifera show different temperature sensitivity, i.e. the relative change in calcite Mg/Ca ratios to changes in temperature (Gussone et al., 2016). The accuracy of Mg/Ca-based temperature estimates in foraminifera species with higher temperature

sensitivity (*Uvigerina peregrina* > *Cibicidoides spp*.) is thus more impacted by contamination. *Cibicidoides spp*. has previously been shown to have different temperature sensitivity at different temperature ranges (Elderfield et al., 2006). Temperature sensitivity of *Cibicidoides spp*. including *Cibicidoides mundulus* and *Cibicidoides wuellerstorfi* is higher at temperatures above 3˚C and therefore the relative impact is stronger in temperatures above 3˚C.

## 4.9. Different contamination thresholds

Different studies have used different indicators and thresholds to monitor silicate contamination and Mn-oxide coatings. Barker et al. (2003) consider correlation between Fe/Ca, Al/Ca ratios with Mg/Ca ratios as indicator of silicate contamination. Elderfield et al. (2010) have used contamination thresholds of 0.4, 0.1 and 0.1 mmol mol[-1] of Al/Ca, Fe/Ca and Mn/Ca ratios respectively as indicator of contamination but also states because of difficulties with precision of Al concentrations, the Mg/Ca ratios were not excluded based on high Al/Ca ratios alone. Yu and Elderfield (2008) used correlation between Al/Ca and Mn/Ca

ratios with Mg/Ca ratios to assess contamination. Capelli et al. (2005) have used Al/Ca ratios <1 mmol mol[-1] and correlation with Mg/Ca ratios to identify silicate contamination. In contrast, Stripe et al. (2021) have used more strict thresholds of 0.0952, 0.0296 and 0.0189 µmol mol[-1] of Al/Ca, Fe/Ca and Mn/Ca ratios, respectively. While the most common contamination is based on correlations with Fe/Ca, Al/Ca and Mn/Ca, the outlined differences cause uncertainty when comparing results between studies. When only using correlations between Fe/Ca, Al/Ca ratios and Mg/Ca ratios to assess silicate contamination, no

samples of *Cibicidoides spp.* in this study would have been tagged as contaminated. However, when correlations are used in combination with the contamination thresholds, about half of the samples indicate silicate contamination or other contamination. Also, when assessing correlations at species level, i.e. *Cibicidoides wuellerstorfi*, there is a strong correlation between Fe/Ca and Mg/Ca ratios (Figure 1A in Appendix A, excluding *Cibicidoides spp.* and *Cibicidoides mundulus* which were analysed in the same run). The species difference could be due to morphological features of *Cibicidoides wuellerstorfi*

that allow more silicate contaminants to be trapped. On the other hand, if the lower contamination thresholds of Stirpe et al. (2021) are used, all Mg/Ca ratios of this study are suggested to be contaminated with both silicate and Mn-oxide coatings. Correlation between Fe/Ca, Al/Ca and Mn/Ca with Mg/Ca help identify contaminants that also contain Mg (most notably silicate and Mn-oxide coatings) and are therefore relevant for determining calcite Mg/Ca ratios. Excluding samples based on strict contamination thresholds for Fe, Al and Mn, without considering correlations to Mg/Ca ratios risk excluding many

samples that have minor contaminants which do not affect Mg/Ca ratios. These measurements could still prove a reliable estimate of Mg/Ca ratios. Still, any presence of contamination is a concern. Even if it does not produces inaccurate Mg/Ca ratios (in the case that the contaminant does not contain Mg), it introduces uncertainties. The inconsistencies between studies and the uncertainties would be resolved by further examination of appropriate contamination thresholds to be used. Because the degree of contamination effect depends on factors such as average Mg/Ca ratios and temperature sensitivity a more



appropriate contamination threshold should be species specific and at least specific to benthic/planktic since average Mg/Ca

ratios are significantly lower in benthic species (Hasenfratz et al., 2017).

## 5.    SUMMARY AND CONCLUSION

Designed to optimise the relationship between sample cleaning and sample loss during the procedure, in experiments 1-3 varying methanol and ultra-pure water rinses were used and clearly show a substantial effect on the level of silicate

contamination. These experiments showed that the best cleaning method for our study was that of Barker et al. (2003).

The core-top calibration for *Cibicidoides wuellerstorfi* in this study is broadly in line with published data, although there is only one data point in our study at the high temperature end.

Contamination is a general problem. Despite using an established method (Barker et al. 2003), in particular *Uvigerina peregrina* displayed high levels of remanent contamination. The *Uvigerina peregrina* Mg/Ca ratios also indicate that the contamination

indicating thresholds have generally been correct in identifying samples with silicate contamination.

There are several potential sources of error for Mg/Ca ratios including the carbonate ion effect, diagenetic effects, seawater Mg/Ca variability, and vital effects. The main limitation in the use of Mg/Ca as a paleothermometer is a general lack of understanding of benthic foraminiferal Mg incorporation and the relative impact of environmental factors, biogenic controls and diagenetic effects. It is possible that species specific cleaning protocols are needed to improve comparability of data

between studies.

## 6.    Competing Interests

S. Jung is co-editor of the special issue dedicated to Dick Kroon and will not be involved in the handling of this manuscript. The other authors declare that they have no conflict of interest.

## 7.    Author contribution

The research was conceptualised by S. Jung and V. Larsson. V. Larsson designed, carried out experiments, conducted data analysis, visualisation and prepared the original manuscript. This was done with supervision and validation input from S. Jung. S. Jung prepared the final manuscript with input from V. Larsson.

## 8.    Acknowledgments

The authors thank Dr. Laetitia Pichevin for advice on experimental design and statistical analysis, Prof. Raja Ganeshram for

advice on the Mg/Ca cleaning methodology and the PGR staff in the Geoscience Department at The University of Edinburgh for providing administrative help.



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



**Appendix A**

Mg/Ca ratios and contamination indicators

**Table A1** Samples analysed in Experiment 1-3

| Sample ID | Core sample | Size fraction (µm) | Specimens containing | Crushing method |
|---|---|---|---|---|
| *Experiment 1* | | | | |
| 1a | 929, Section 13, 58-58.5 cm | 250-355 | 25 *G. ruber* | Two glass slides |
| 1b | 929, Section 13, 58-58.5 cm | 250-355 | 25 *G. ruber* | Two glass slides |
| 1b | 929, Section 13, 58-58.5 cm | 250-355 | 10 *G. ruber* | Two glass slides |
| 1c | 929, Section 13, 57.5-58 cm | 250-355 | 20 *G. ruber* | Metal pin glass slide |
| 1d | 929, Section 13, 58-58.5 cm | 250-355 | 10 *G. ruber* | Metal pin glass mortar |
| 1e | 929, Section 13, 57.5-58 cm | 250-355 | 13 *G. ruber* | Metal pin glass mortar |
| *Experiment 2* | | | | |
| 2a | 929, Section 13, 29-29.5 cm* | 250-355 | 20 *G. ruber* | Metal pin glass mortar |
| 2b | 929, Section 13, 29-29.5 cm* | 250-355 | 20 *G. ruber* | Metal pin glass mortar |
| 2c | 929, Section 13, 29.5-30 cm* | 250-355 | 20 *G. ruber* | 2-3 s in ultrasound |
| 2d | 929, Section 13, 57.5-58 cm* | 250-355 | 20 *G. ruber* | 2-3 s in ultrasound |
| 2e | 929, Section 13, 58-58.5 cm* | 250-355 | 20 *G. ruber* | Not crushed |
| 2f | 929, Section 13, 57.5-58 cm* | 250-355 | 20 *G. ruber* | Not crushed |
| 2g | 929, Section 13, 57.5-58 cm* | 250-355 | 20 *G. ruber* | Not crushed |
| 2h | 929, Section 13, 57.5-58 cm* | 250-355 | 20 *G. ruber* | Not crushed |
| *Experiment 3* | | | | |
| 3a | 929, Section 14, 12.5-13.5 cm* | 250-355 | 20 *G. ruber* | Metal pin glass mortar |
| 3b | 929, Section 14, 17-17.5 cm* | 250-355 | 20 *G. ruber* | Metal pin glass mortar |
| 3c | 929, Section 14, 105.5-106 cm | 250-355 | 10 *G. ruber* | Metal pin glass mortar |
| 3d | 929, Section 13, 105.5-106 cm | 250-355 | 50 *G. ruber* | Metal pin glass mortar |
| 3e | 929, Section 13, 105.5-106 cm | 250-355 | 30 *G. ruber* | Metal pin glass mortar |
| 3f | 929, Section 13, 105.5-106 cm | 250-355 | 20 *G. ruber* | Metal pin glass mortar |
| 3g | PE303-13[B], CT, 0-1 cm | 250 - >450 | 5 *Cib. spp.* | Metal pin glass mortar |
| 3h | PE303-17[A], CT, 0-1 cm | 250 - 450 | 5 *Cib. spp.* | Not crushed |
| 3i | PE303-17[A], CT, 0-1 cm | 250 - 450 | 10 *Uvi. spp* | Metal pin glass mortar |

*used for comparison to previously measured Mg/Ca ratios by Saher et al. (2009)



**Table A2.** Results from Experiment 1-3

| Sample ID | Species | Mg/Ca (mmol/mol) | Fe/Ca (mmol/mol) | Al/Ca (mmol/mol) | Mn/Ca (mmol/mol) | Ca (ppm) | Note |
|---|---|---|---|---|---|---|---|
| 1a | *G. ruber* | 35.23 | <LOD | <LOD | | 0.55 | |
| 1b | *G. ruber* | 3.89 | <LOD | <LOD | | 7.50 | |
| 1c | *G. ruber* | 5.18 | <LOD | <LOD | | 3.75 | |
| 1d | *G. ruber* | 5.84 | <LOD | <LOD | | 1.68 | |
| 1e | *G. ruber* | 4.54 | <LOD | <LOD | | 5.10 | |
| 1f | *G. ruber* | 5.70 | <LOD | <LOD | | 5.66 | |
| 2a | *G. ruber* | 5.64 | 0.51 | 2.36 | 0.18 | 19.33 | |
| 2b | *G. ruber* | 4.40 | <LOD | 3.80 | 0.15 | 7.92 | |
| 2c | *G. ruber* | 4.66 | 0.18 | 1.84 | 0.17 | 22.54 | |
| 2d | *G. ruber* | 6.58 | 0.78 | 3.61 | 0.20 | 13.99 | |
| 2e | *G. ruber* | 4.71 | 0.27 | 1.67 | 0.15 | 30.16 | |
| 2f | *G. ruber* | 5.49 | <LOD | 4.34 | 0.16 | 7.32 | |
| 2g | *G. ruber* | 4.85 | 0.43 | 2.11 | 0.18 | 24.18 | |
| 2h | *G. ruber* | 9.41 | 1.56 | 3.96 | 0.34 | 30.92 | |
| 3a | *G. ruber* | 3.17 | 0.39 | 0.91 | 0.11 | 2.15 | |
| 3b | *G. ruber* | 3.7 | <LOD | <LOD | 0.13 | 21.15 | |
| 3c | *G. ruber* | 3.99 | 1.08 | 0.96 | 0.17 | 27.48 | |
| 3d | *G. ruber* | 3.59 | <LOD | 0.87 | 0.15 | 11.27 | |
| 3e | *G. ruber* | 4.48 | 0.56 | 1.61 | 0.18 | 88.35 | |
| 3f | *G. ruber* | 3.78 | 0.39 | 5.93 | 0.14 | 31.66 | |
| 3g | *G. ruber* | 2.37 | <LOD | 3.36 | 0.17 | 35.39 | |
| 3h | *G. ruber* | 2.54 | 1.08 | 0.91 | 0.04 | 4.32 | |
| 3i | *G. ruber* | 2.75 | <LOD | <LOD | 0.07 | 11.80 | |






**Table A3.** Mg/Ca, Ca and contamination indicators (Fe/Ca, Al/Ca and Mn/Ca) of samples containing *G. ruber* analysed in the same runs alongside core-top samples.

| Core | Species | Mg/Ca | Fe/Ca | Al/Ca | Mn/Ca | Ca | Note* |
|------|---------|-------|-------|-------|-------|------|-------|
| NIOP929 | *G. ruber* | 3.70 | 0.14 | 0.36 | 0.17 | 17.82 | *C. spp.* |
| NIOP929 | *G. ruber* | 3.56 | 0.10 | 0.35 | 0.19 | 25.60 | *C. spp.* |
| NIOP929 | *G. ruber* | 3.17 | 0.09 | 0.31 | 0.12 | 28.36 | *C. spp.* |
| NIOP929 | *G. ruber* | 3.16 | 0.23 | <LOD | 0.17 | 10.60 | *C. spp.* |
| NIOP929 | *G. ruber* | 4.18 | 0.12 | 0.29 | 0.17 | 19.78 | *C. spp.* |
| NIOP929 | *G. ruber* | 3.26 | 0.10 | 0.25 | 0.15 | 24.51 | *C. spp.* |
| NIOP929 | *G. ruber* | 5.22 | 0.38 | 1.31 | 0.11 | 24.48 | *U. peregrina* |
| NIOP929 | *G. ruber* | 4.77 | 0.24 | 0.92 | 0.14 | 50.88 | *U. peregrina* |
| NIOP929 | *G. ruber* | 5.1 | 0.30 | 1.04 | 0.12 | 39.95 | *U. peregrina* |

*Analysed in run alongside *C. spp*/*U. peregrina*

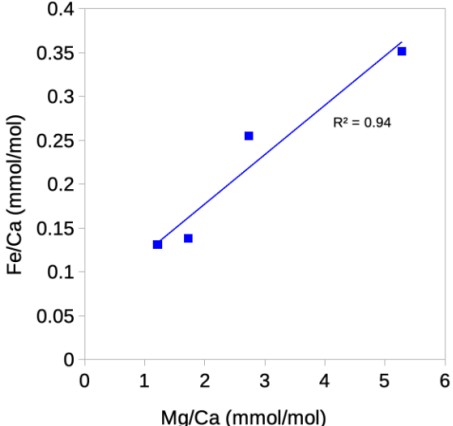

**Fig. A1** Correlation between Fe/Ca ratios and Mg/Ca ratios in Cibicidoides wuellerestorfi. One datapoint is excluded Fe below limit of detection.




**Appendix B**

ICP-OES calibration curves and standards

**Table B1** Concentration of standards used (Mg in ppb and Ca in ppm) in Mg/Ca analysis for calibration and for matrix effect.

| Tube | Sample Labels | Al 396.152 mg/L | Ca 315.887 mg/L | Ca 317.933 mg/L | Ca 422.673 mg/L | Mg 279.553 ug/L | Mg 280.270 ug/L | Mg 285.213 ug/L |
|------|---------------|-----------------|-----------------|-----------------|-----------------|-----------------|-----------------|-----------------|
| 1 : 1 | UoE benthos Blank | 0.000000 | 0.000000 | 0.000000 | 0.000000 | 0.000000 | 0.000000 | 0.000000 |
| 1 : 2 | UoE benthos St1 | | 13.8500 | 13.8500 | 13.8500 | 75.5000 | 75.5000 | 75.5000 |
| 1 : 3 | UoE benthos St2 | | 16.5000 | 16.5000 | 16.5000 | 258.000 | 258.000 | 258.000 |
| 1 : 4 | UoE benthos St3 | | 13.8500 | 13.8500 | 13.8500 | 339.000 | 339.000 | 339.000 |
| 1 : 5 | UoE benthos St4 | | 13.8500 | 13.8500 | 13.8500 | 466.000 | 466.000 | 466.000 |
| 1 : 6 | UoE benthos St5 | | 13.8500 | 13.8500 | 13.8500 | 677.000 | 677.000 | 677.000 |
| 1 : 7 | UoE benthos St6 | | 13.8500 | 13.8500 | 13.8500 | 1379.00 | 1379.00 | 1379.00 |
| 1 : 8 | Standard 8 | | | | | | | |
| 1 : 9 | Standard 9 | 0.500000 | 0.500000 | 0.500000 | 0.500000 | | | |
| 1 : 10 | Standard 10 | 2.50000 | 2.50000 | 2.50000 | 2.50000 | | | |
| 1 : 11 | Standard 11 | 10.0000 | 10.0000 | 10.0000 | 10.0000 | | | |
| 1 : 12 | Standard 12 | 50.0000 | 50.0000 | 50.0000 | 50.0000 | | | |


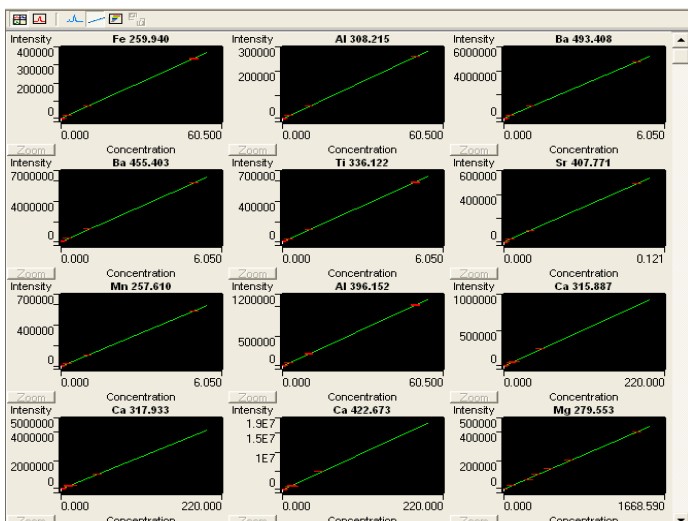

**Fig. B1**. Print screen of ICP Expert showing calibration curves for standards in *Uvigerina peregrina* analysis

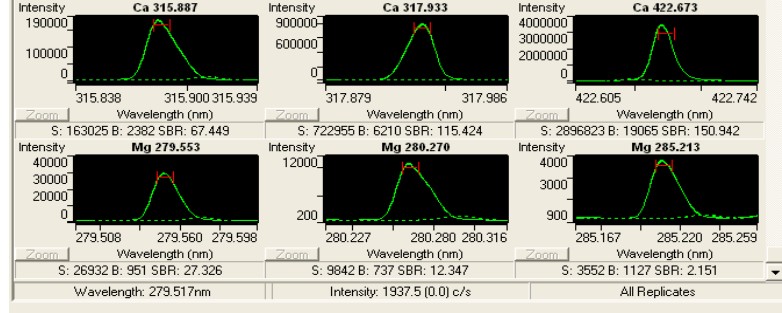

**Fig. B2.** Print screen of ICP Expert showing intensity curves in *Cibicidoides spp* analysis




**Appendix C**

Qualitative observations of benthic foraminifera samples analysed

**Table 1C** Images and qualitative observations of 250-450 µm size fractions of core top samples containing benthic foraminifera
tests analysed.

| | | |
|---|---|---|
| PE303-3 | PE303-4c | PE303-6c |
| - no debris or quartz<br><br>- clean<br><br>- broken fragments present but most tests intact<br><br>- mostly unstained/ minor discoloration<br><br>- no sign of secondary calcification or corrosion | - 50% quartz<br><br>- clean<br><br>- tests mostly intact<br><br>- mostly unstained/ minor discoloration<br><br>- no sign of secondary calcification or corrosion | - 50% quartz<br><br>- clean<br><br>- tests mostly intact<br><br>- mostly unstained/minor discoloration<br><br>- no sign of secondary calcification or corrosion |
| PE303-14a | PE303-17a | PE303-18b |



| - no debris or quartz<br><br>- broken tests present but most intact<br><br>- some signs of secondary calcification and corrosion white no discoloration | - no debris or quartz<br><br>- mixed condition, visible mud on tests<br><br>- discoloration and corrosion | - debris and quartz present<br><br>- visible mud on tests<br><br>- broken fragments<br><br>- discoloration and corrosion present<br><br>- no sign of secondary calcification |
|---|---|---|
| | | |
| PE303-22 | PE303-50 | PE304-9 |
| - debris present, no quartz<br><br>- mixed condition, some discoloration corrosion and secondary calcification<br><br>- broken fragments | - >50% quartz, no debris<br><br>- clean<br><br>- mixed condition, minor discoloration and some visible mud<br><br>- some broken tests<br><br>- minor corrosion<br><br>- no sign of secondary calcification | - debris and quartz present<br><br>- visible mud<br><br>- broken fragments >50%<br><br>- discoloration<br><br>- corrosion<br><br>- no sign of secondary calcification |
| | | |





| PE304-25 | PE304-30 | |
|---|---|---|
| - no quartz or debris<br><br>- most tests have visible mud<br><br>- broken fragments<br><br>- brown/orange discoloration<br><br>- no visible sign of corrosion or secondary calcification | - no quartz or debris<br><br>- mixed condition, some visible dirt<br><br>- minor discoloration<br><br>- broken test fragments<br><br>- corrosion<br><br>- minor secondary calcification | |