# Peer review of "Western Indian Ocean bottom water temperature calibration – are benthic foraminifera Mg/Ca ratios a reliable palaeothermometry proxy?"

_EGUsphere, 2024_

## Referee Comment (RC1)

Review:

"Western Indian Ocean bottom water temperature calibration – are benthic foraminifera Mg/Ca ratios a reliable palaeothermometry proxy?"

Larsson & Jung, 2024

The manuscript provides a new dataset of various element/Ca ratios, focussing on Mg/Ca ratios, for three benthic foraminifera species. The data originate from the western Indian Ocean and were checked by the authors for their applicability as a paleothermometer proxy for bottom water temperatures. Although the number of samples seems too small to establish a valid new calibration, the manuscript contributes to an improved understanding of benthic foraminifera and their usability as palaeoproxy in this area.

Furthermore, the authors compare their data set with two others from the region and evaluate the quality and relevance of their data as a palaeoproxy in the discussion chapter, under consideration of significant literature. As I myself am not particularly familiar with benthic foraminifera, I cannot make a qualitative statement about the methodology which is used here.

The authors have presented their results in a sufficient number of graphics, although some of them still need a bit of reworking. Furthermore, the language is a little clumsy in some places, but as I am not a native speaker myself, I have only made minor comments here.

Overall, I rate the manuscript as good and recommend publication, although there are still a few points that need to be improved.

Be more consistent throughout the manuscript e.g.:

1) Line 172 "(Table A1 in Appendix A)" vs. Line 201 "(Appendix A Table A1)"

2) Abbreviation of the foraminifera. Sometimes you wrote *Cibicidoides mundulus* vs *C. mundulus*. You can write the entire name throughout the manuscript, but I recommend to write the full genus and species name when you first mention it and then continue throughout the manuscript with the abbreviation.

3) Same for figure vs. fig. vs Figure (and table, Table, tab.), choose one and stay consistent.

Figures and Tables

Figure 2 would benefit from gridding the potential temperature.

Figure 4: Use either two different symbols or two different colors (best is both), it is hard to distinguish between both.

Figure 6: Same like Figure 4. I find it hard to distinguish between the both triangles, since the dark blue and black do not show a strong contrast, but also the other blue symbols could benefit from a different color. Also, the graphic labelling is confusing here. I would write "Mg/Ca ratio against (a) Fe/Ca ratios, (b) Al/Ca ratio and (c) Mn/Ca ratio in *Cibicidoides spp.* …"

Figure 9: Again. Maybe use a circle for *C. spp.* ?

Table 1: "*Wuellerstrofi*" is written in capital letters, change it.

Table A2: Numbers e.g. at 2a, 2g and 3a seem to be smaller.

Table A2: Why is the species at samples from 3g to 3i *G. ruber*? I thought it is *Uvigerina spp.* and *Cibicidoides spp.*

Minor revisions:

Line 74: One space too many after "i.e."

Line 92: strange bracket after "(Stripe et al., 2021)" and one space too many

Line 107-109: You say "three" water masses but listed just "two" of them

Line 136: two dots at the end of the sentence

Line 138: One space too many after "(Table 1)"

Line 143: first mention of *G. ruber* -> write the entire name

Line 153: write „In experiment 1" to keep it consistent. Furthermore, mention that you used 6 sets with a varying amount of *G. ruber* somewhere.

Line 162-164: 5.81 ppm and 9.53 ppm, where does these numbers come from? In Table A2 I see three tests with *G. ruber* for the crushing experiment between two glass slides (Ca 0.55, 7.5 and 3.75 ppm = **3.93**) and three tests with *G. ruber* for the crushing experiment using a metal pin (1.68, 5.1, 5.66 = **4.15**). Same for Mg/Ca, can't find the 3.43 and 3.53 mmol mol-1 in A2, especially since sample 1a has incredibly high values of 35.23 mmol mol-1.

Line 170: maybe add "(0.55 to 7.50 ppm in both crushing between two glass slides and when using a metal pin, Table 1A and A2 in Appendix A)" to make sure that the general results, regardless of the method used, are really low.

Line 180-184: For your average values 0.38, 2.91 and 3.01 mmol mol-1, I have 0.37, 2.90 and 3.02 respectively. I suppose this is because you used a higher number of decimals in your original calculation. I am mention this in case you want to change it, but I think this is fine. For your mean Fe/Ca ratio of 0.61 mmol mol-1, I have 0.57 mmol mol-1 instead. Please check this again and correct it.

Line 199: (6x25): change sentence: "In the procedure, specimens of *Globigerinoides ruber* (6 sets with a varying amount of 10 - 50 individuals; Table A1, Appendix) picked from …"

Line 200: remove point after "*Uvigerina…"*

Line 201: (Table A1 in Appendix A) -> see Line 172

Line 214: Make it two sentences: "… proposed by Hasenfratz et al. (2017). This suggest that Mn-oxide…"

Line 228 230: change to (Figure B1 in Appendix B)

Line 262: One space too many

Line 280: Table A3?

Line 299-300: I see **four** samples in water depth deeper than 2500 m and **five** samples in water depth <1500 m. Furthermore, instead of "Below 2500 m" maybe write "In water depth >2500 m …"

Line 303: *Cibicidoides* in italics

Line 308 – 310: rephrase sentence: „The Mg/Ca ratios of *Cibicidoides spp.*, although higher in their Fe/Ca ratios than >0.1 mmol mol-1, were also included, since they show no correlation between Mg/Ca ratios and Fe/Ca ratios."

Line 310: One space too many between "Table" and "2". Also, write "Figure 6" in capital letters to stay consistent.

Line 312: *C. mundulus* and *C. wuellerstorfi*, stay consistent.

Line 313: "Figure 9". Also, remove point after "*Cibicidoides*". Richtig: *Cibicidoides spp.*

Line 318: "Figure 9"

Line 314 & 320: BWT

Line 321: Sentence in parentheses not in italics

Line 322: "Figure 9"

Line 331: *Cibicidoides spp.*

Line 337: *Cibicidoides spp.*; Also, one space too many between "Table 2" and "When"

Line 353: One space too many between "Table 2" and "It"

Line 356: compared to what? Samples from *Cibicidoides spp.*?

Line 359: One space too many between "ratio" and "The"

Line 366: "Figure 10"

Line 369: "Figure 10"

Line 398: "Figure 11"

Line 400: space

Line 401: "SW Indian Ocean"

Line 426: Bracket after Figure 6 missing

Line 427: Bracket in the wrong place

Line 430 - 432: rephrase sentence e.g.: "Although the leaning procedure by Barker et al. (2003) has been widely used (e.g., …) the removal of Mn-Mg coatings is still inefficient (Hasenfratz et al., …)."

Line 439 – 441: rephrase sentence e.g.: "The high Fe/Ca ratios as well as the high Al/Ca ratios in most samples of all species used here (Table 2) indicate inefficient removal of silicate contaminants, suggesting that the number of rinse/ultrasonication repetitions of the Barker et al. (2003) procedure is inadequate."

Line 449: what is with the value of 0.15 mmol mol-1 in Table 2 for the lowest range of *Uvigerina peregrina*?

Line 450: add "(… 0.35 mmol mol-1; Table 2)"

Line 451: rephrase term: compared to Fe/Ca rations in *Cibicidoides wuellerstorfi* below 0.04 mmol mol-1.

Line 455/457: Do you mean Table A3?

Line 457: add "(0.13 and 0.31 mmol mol-1)" behind *Cibicidoides spp.* -> There is also a dot missing after "*spp*"

Line 465: write: "… core depth, water depth, and morphology"

Line 474-476: Don't understand this sentence. What is a nearby region here?

Line 477: I think there is a comma missing between contamination and Fe/Ca …?

Line 478: Bracket closed after "Figure 8"

Line 479: Bracket closed after "Figure 7"

Line 498: Missing commas before and after "respectively", as well as before "but"

Line 507: one space too many between "i.e." and "*Cib*"

---

## Referee Comment (RC2)

EGUsphere – May 2024
**Western Indian Ocean bottom water temperature calibration – are benthic foraminifera Mg/Ca ratios a reliable paleothermometery proxy?**
**Larsson, V. & Jung, S.**

**Overall Review:**

This study presents some new Mg/Ca (and other elemental/Ca data) from benthic foram samples from core-top locations in the Western Indian Ocean basin and compares it to previously reported values from other locations in the wider Indian Ocean and beyond. They walk through three experimental methods for cleaning prior to Mg/Ca analysis and compare results in the context of contamination and data exclusionary procedures. They show agreement with some previously published calibrations for Mg/Ca – Bottom Water Temperature. This is a significant amount of work and the authors do have some exciting things to share.

However, their data is quite limited after removal of samples with apparent contamination, such that their final calibration model only includes 4 samples. As far as a method-based paper discussing the range of cleaning methods and implications of those methods on data collection, this paper is valuable. As far as a new model for reconstructing temperatures, this data should be taken with caution due to such low final sample numbers. However, their data do seem to fall within previously published values which shows good continuity. They could stress this more in the abstract as well, and make it very clear that their new calibration model contains only n = 4 samples.

I believe the paper should be published, but first with edits. Specifically, some of the text is confusing to follow as written, with many subsections in the discussion that are quite disparate/not well connected. If the authors are able, it would be good to go back through the discussion in particular to draw out and make very clear the major findings of their work. In general, a clearer outline of what this study brings to the literature/scientific field and the suggestions/findings for future work is needed. The Summary and Conclusions section is quite sparse in this regard and could be used as a place to discuss their findings in greater depth.

With respect to the methodological testing (cleaning methods), it would be helpful if the authors can work to make it far clearer what the difference between experiment 1 cleaning versus experiment 2 versus experiment 3 cleaning methods were – it's difficult the way it is worded now to follow. Perhaps a table outlining the differences of each procedure, and which steps were included in each so people can easily cross compare the methods would be instructive. Is the only difference that the methanol washes timing is reduced from 1-2 min to 20 seconds? Is it expected that the cleaning procedure would work the exact same way on the benthic species? Can the authors make a note of their expectations on this and reasoning for using *G. ruber* as their cleaning methodology species?

Please also update figures throughout to make it clearer for the reader to distinguish between datapoints, as they stand now many of the symbols are too small and soo timilar to one another.

Overall, it is difficult to determine whether they conclude that benthic foraminifera Mg/Ca ratios are a reliable paleo thermometer in the Indian Ocean. Please be more explicit in your conclusions about your findings and how they fit within the larger context of benthic foram Mg/Ca paleothermometery.

**Detailed Review:**

**Figure edits:**

**Figure 1 –** maybe label with (a) and (b) so that it's clear beyond colour which map is showing which core-tops.

**Figure 2 –** label with (a) and (b) (c) so it's clear in your figure caption when you are speaking about temperature and salinity and the transect map.

**Figure 3 –** Be very clear in your figure caption. Indicate when you are referring to the 'blue arrow' and 'red arrow' because readers may think you are referring to the water temperature blue and red colours.

**Figure 4 –** Perhaps it might be easier to tell the difference between crushed versus not crushed samples if you colour them differently or have one filled in one empty triangles?
Also - does it really make sense to have one regression line that includes both crushed and un-crushed data points - what about two separate regressions showing the relationships across treatments?
Figure 4 caption: Is there a way to quickly describe the procedure here instead of referring to it as from Experiment 2? perhaps you can say using the method with shorter methanol cleaning step?

**Figure 6 -** In all species evaluated across all cores? indicate this if that's the case. Otherwise, where (which cores) are these data from?

Also – panel D doesn't exist – where you indicate in your caption there should be the correlation between total Ca and contamination (Fe/Ca)

Rather than just saying 'horizontal lines' say "Orange lines show..." and indicate which value corresponds to which Element/Ca ratio.

**Figure 8 –** The legend box is outside of the figures, try to align so it does not cross the figure border the way it does now.

**Figure 9 –** How many samples make up each relationship – indicate sample number with " n = "
Capitalize the C. for C. wuellerstorfi on figure
On line 329: should also include (u) here after Mg/Ca for (uncontaminated) indicator you have in your figure legend.

**Figure 10 -** So all calibration models/ data presented are from this one species? make sure that is clear in the text above
Include reference/citation for the S. W Indian Ocean grey samples in your figure legend as you do for the purple and blue.

**Line by line edits:**

**14:** the word entailed doesn't seem appropriate. Perhaps the word elucidated?

**19:** with what error can the 'wider' Indian ocean calibrations be used in the Western Indian Ocean? Could report that error here for brevity and maximal impact of abstract

**22:** With what error can BTW be reconstructed? What does your calibration error translate to in degrees C error of the BTW?

**24:** remove the word 'the' after controls…

**25:** the re-distribution of heat **in the oceans** is an ….

**28:** sentence should read: "… of the sensitivity and changes in thermohaline circulation. For example, on glacial-interglacial time scales, NADW and AABW…"

**34:** can use $\delta^{18}O$ here instead of 'stable oxygen isotopes because you've already defined it earlier

**41:** perhaps instead of "being developed" saying "is still unresolved" is better?

**43:** Before starting in on the discussion of Mg incorporation, make clear that forams make calcite tests which is a calcium carbonate matrix… and describe that Mg substitutes for Ca in the lattice.

**43:** remove the word "also" after "Mg2+ are also incorporated…"

**44:** indicate that Mg/Ca does not just depend on Mg/Ca of seawater and elemental partitioning…it's an endothermic process that relies on water temperature – hence the reason we can use it to reconstruct water temperatures!

**48:** Define BWT here before using the acronym as it has not yet been defined in the text.

**49:** Temperature appears to be the dominant environmental factor driving what? Indicate you mean the incorporation of Mg?

**52:** carbonate ion saturation being dominant of/at what? What process is it affecting? Mg incorporation? explain

**54:** What do you mean by 'spatially-varying' please elaborate here.

**71:** "larger lowering" is awkward wording, consider rephrasing this

**80:** ".. determining **bulk or whole specimen** calcite Mg/Ca ratios"

**84:** "*Cibicidoides wuellerstorfi* **has** been one of…"

**85:** provide example citations for the use of benthic species for stable $\delta^{18}O$ and $\delta^{13}C$ reconstructions.

**87:** when referring to Mg/Ca incorporation – it is better to say Mg/Ca signatures OR Mg incorporation.

**91:** "… being usable…" is an awkward way of saying this. Perhaps "employed"

**92:** "Uncertainties remain (Stirpe et al. 2021)" – be clearer here, what uncertainties are you referring to that are raised in Stirpe et al. 2021.

**92:** "… entailing the need…" entailing is an awkward word choice: perhaps 'pointing to the need' or 'elucidating the need'

**94:** remove the word help before 'improving' and change 'improving' to 'improve'

**95/96:** you compare to calibrations from the Indian Ocean – are these previously published calibrations? If so list the citations if they are not too lengthy?

**108:** should say… "…core-top transect is comprised of…"

**108:** you only indicate 2 water masses in this first list… and go on to introduce the third later but it is slightly confusing/misleading to say three here and not list all three

**113:** where are the CTD temperature data from? When were they collected relative to the core-top collections?

**134:** no need for brackets around "where possible" when you are using commas

**144:** NIOP929 – what is this? Which core? From what cruise/expedition/researchers?

**145:** The sentence should read: "The samples were wet sieved over…"

**147:** The sentence should read: "....remove silicates, a hydrogen peroxide treatment to remove organic matter, and followed by an acid ..."

**149:** "depend" should be "depends"

**153:** The sentence should read: "…except for a reduction in duration of the methanol washes (25 seconds…"

**154:** The language "following previously analysed samples in the laboratory." Is not clear – please clarify and/or rephrase.

**159:** for how long was the hydrogen peroxide treatment? At what temperature?

**166:** Does this also mean you lose Mg in the glass slide method as well as Ca in order to keep the ratios similar to the pin method this must be true?

**178:** If the data in Figure 4 below show experiment 2 results, then pointing to it here for experiment 1 calcium values doesn't make sense

**183:** Did you run a sensitivity test/ leverage test to see how much leverage that outlier has on your regression/model? if it's low then you can likely include it, if it's high then it's likely skewing the results.

**184:** Is it really true that there is no relationship between Al/Ca and Mg/Ca? or is it just non-linear like logarithmic? Looks like it would be.

**185:** The correlation for not crushed is only really strong if the outlier is included, otherwise just clusters... the correlation for crushed tests is linear however

**224:** Please describe in more detail what the standards were - you should have that information from the lab in which these samples were run and it's standard practice to include the type/name of the specific standards used.

**225:** You should rename this Figure B1 - to indicate you are referring to Figure 1 of Appendix B and not Figure 1 of the manuscript.

**229:** 'effect' should be 'effects'

**Section headings 3.1 and 3.2** - you refer to more than just Mg/Ca ratios in these sections, perhaps re-naming to say elemental ratios or something similar?

**261-264:** It does not look like the samples are below the contamination thresholds? you even indicate in table 2 which samples are eliminated due to contamination.... so this statement can't be true.

**298:** $r^2$ is a different metric from $R^2$ – you have used both throughout the text, so go through and ensure

**310:** Perhaps a sentence following indicating how many samples remained for the core-top calibration is useful here.

**318:** capitalize the 'f' in figure 9 – Figure 9

**323:** This calibration is based on 4 samples, one of which is quite far away from the others - could be contributing significant leverage to the calibration... i.e. without it the entire relationship would fall away. Have you tested this? Also, if you are going to report this calibration model (or any throughout the manuscript) it is good practice to indicate the number of samples/data points you have that built the model.. in this case: n = 4.

**348:** Maybe rather than 'abnormally' say anomalously high, not abnormally since you have cited evidence where similar reports have been made

**350-354:** So did you exclude this sample from calibration models? make it very clear if so.

**355- 360:** So did you exclude this sample from calibration models? make it very clear if so.

**362:** Are these studies also reporting only c. wuellerstorfi calibration models or which species are they reporting? indicate this here if they are species-specific or mixed-species models!

**369:** you have ??? after Figure in your reference to figure 11

**393:** the word maybe should be two words in this context: may be

**426:** ) close (Figure 6) with a bracket

**428:** under what subsection/subtitle number are you pointing readers to for discussion on degree of contamination?

**474:** This first sentence is awkward wording...do you mean to say your core top samples are closely located to previously published data?

**514:** risk should be risks

**517-518:** How? if you are to suggest more work should be done it's great if you can point to examples of next steps

**520:** benthic/planktic should be benthic versus planktic

**526-527:** Also include that this calibration only includes 4 samples.

**534-535:** It would be great to say something more about your study.... while you are not able to speak on the other uncertainties/limiting factors, you have shown here that cleaning procedures may need to be refined and conducted species-by-species.

---

## Author Comment (AC1)

RC1:

Review:

"Western Indian Ocean bottom water temperature calibration – are benthic foraminifera Mg/Ca ratios a reliable palaeothermometry proxy?" Larsson & Jung, 2024

The manuscript provides a new dataset of various element/Ca ratios, focusing on Mg/Ca ra- tios, for three benthic foraminifera species. The data originate from the western Indian Ocean and were checked by the authors for their applicability as a paleothermometer proxy for bot- tom water temperatures. Although the number of samples seems too small to establish a valid new calibration, the manuscript contributes to an improved understanding of benthic forami- nifera and their usability as palaeoproxy in this area.

Furthermore, the authors compare their data set with two others from the region and eval- uate the quality and relevance of their data as a palaeoproxy in the discussion chapter, under consideration of significant literature. As I myself am not particularly familiar with benthic foraminifera, I cannot make a qualitative statement about the methodology which is used here.

The authors have presented their results in a sufficient number of graphics, although some of them still need a bit of reworking. Furthermore, the language is a little clumsy in some places, but as I am not a native speaker myself, I have only made minor comments here.

Overall, I rate the manuscript as good and recommend publication, although there are still a few points that need to be improved.

*Thank you for carefully reviewing our paper and highlighting the inconsistencies.*

Be more consistent throughout the manuscript e.g.:

1) Line 172 "(Table A1 in Appendix A)" vs. Line 201 "(Appendix A Table A1)"

2) Abbreviation of the foraminifera. Sometimes you wrote *Cibicidoides mundulus* vs *C. mundu- lus*. You can write the entire name throughout the manuscript, but I recommend to write the full genus and species name when you first mention it and then continue throughout the manuscript with the abbreviation.

3) Same for figure vs. fig. vs Figure (and table, Table, tab.), choose one and stay consistent.

*We thank the reviewer for these comments. Inconsistencies will be reviewed, as suggested, in the final submission.*

Figures and Tables

Figure 2 would benefit from gridding the potential temperature.

*Figure 2 – We appreciate this comment and will add a version with grided displays of temperature and salinity.*

Figure 4: Use either two different symbols or two different colors (best is both), it is hard to distinguish between both.

*Figure 4 – two different symbols have been used but we agree that using both different colours and symbols would clarify it further. Please see comment in relation to figure 9.*

Figure 6: Same like Figure 4. I find it hard to distinguish between the both triangles, since the dark blue and black do not show a strong contrast, but also the other blue symbols could ben- efit from a different color. Also, the graphic labelling is confusing here. I would write "Mg/Ca ratio against (a) Fe/Ca ratios, (b) Al/Ca ratio and (c) Mn/Ca ratio in *Cibicidoides spp.* …"

*Figure 6 – appreciate your input on this. We will revise the coloring used in this figure. Please also see comment in figure 9.*

Figure 9: Again. Maybe use a circle for *C. spp*. ?

*Figure 9 – We will reassess all figures for both consistency in symbols as well as the use of colour.*

Table 1: "*Wuellerstrofi*" is written in capital letters, change it.

*Table 1 – Thanks for spotting this. We will correct this.*

Table A2: Numbers e.g. at 2a, 2g and 3a seem to be smaller.

*Table A2 – Thanks for spotting this. We will correct this.*

Table A2: Why is the species at samples from 3g to 3i *G. ruber*? I thought it is *Uvigerina spp.* and *Cibicidoides spp.*

*Table A2 – Correct, thanks for noticing. That is indeed a mistake, it will be corrected.*

Minor revisions:

*We thank the reviewer for the very thorough assessment of our manuscript. In relation to the minor revisions comments, we will address all purely editorial issues raised in the list below and correct the text accordingly. Additional comments are added below.*

Line 74: One space too many after "i.e."

Line 92: strange bracket after "(Stripe et al., 2021)" and one space too many Line 107-109: You say "three" water masses but listed just "two" of them Line 136: two dots at the end of the sentence

Line 138: One space too many after "(Table 1)"

Line 143: first mention of *G. ruber* -> write the entire name

Line 153: write „In experiment 1" to keep it consistent. Furthermore, mention that you used 6 sets with a varying amount of *G. ruber* somewhere.

Line 162-164: 5.81 ppm and 9.53 ppm, where does these numbers come from? In Table A2 I see three tests with *G. ruber* for the crushing experiment between two glass slides (Ca 0.55, 7.5 and 3.75 ppm = **3.93**) and three tests with *G. ruber* for the crushing experiment using a metal pin (1.68, 5.1, 5.66 = **4.15**). Same for Mg/Ca, can't find the 3.43 and 3.53 mmol mol-1 in A2, especially since sample 1a has incredibly high values of 35.23 mmol mol-1.

*We will ensure that tables in the final version contain all data.*

Line 170: maybe add "(0.55 to 7.50 ppm in both crushing between two glass slides and when using a metal pin, Table 1A and A2 in Appendix A)" to make sure that the general results, re- gardless of the method used, are really low.

*Thanks for this suggestion.*

Line 180-184: For your average values 0.38, 2.91 and 3.01 mmol mol-1, I have 0.37, 2.90 and 3.02 respectively. I suppose this is because you used a higher number of decimals in your orig- inal calculation. I am mention this in case you want to change it, but I think this is fine. For your mean Fe/Ca ratio of 0.61 mmol mol-1, I have 0.57 mmol mol-1 instead. Please check this again and correct it.

*Yes, the difference is a result of different decimals in the original calculation.  Thanks for pointing this out. We will check this.*

Line 199: (6x25): change sentence: "In the procedure, specimens of *Globigerinoides ruber* (6 sets with a varying amount of 10 - 50 individuals; Table A1, Appendix) picked from …"

Line 200: remove point after "*Uvigerina…*"

Line 201: (Table A1 in Appendix A) -> see Line 172

Line 214: Make it two sentences: "… proposed by Hasenfratz et al. (2017). This suggest that Mn-oxide…"

Line 228 230: change to (Figure B1 in Appendix
B) Line 262: One space too many

Line 280: Table A3?

*Thanks for noting this issue. Figure and table referencing will be corrected where needed in the final version.*

Line 299-300: I see **four** samples in water depth deeper than 2500 m and **five** samples in water depth <1500 m. Furthermore, instead of "Below 2500 m" maybe write "In water  depth

>2500 m …"

*Yes correct, this is a mistake. It will be corrected in the final version. Thanks for noticing this.*

Line 303: *Cibicidoides* in italics

Line 308 – 310: rephrase sentence: „The Mg/Ca ratios of *Cibicidoides spp.*, although higher in their Fe/Ca ratios than >0.1 mmol mol-1, were also included, since they show no correlation between Mg/Ca ratios and Fe/Ca ratios."

*Will do.*

Line 310: One space too many between "Table" and "2". Also, write "Figure 6" in capital letters to stay consistent.

Line 312: *C. mundulus* and *C. wuellerstorfi*, stay consistent.

Line 313: "Figure 9". Also, remove point after "*Cibicidoides*". Richtig: *Cibicidoides spp.*

Line 318: "Figure 9"

Line 314 & 320: BWT

Line 321: Sentence in parentheses not in italics
Line 322: "Figure 9"

Line 331: *Cibicidoides spp.*

Line 337: *Cibicidoides spp.*; Also, one space too many between "Table 2" and "When"
Line 353: One space too many between "Table 2" and "It"

Line 356: compared to what? Samples from *Cibicidoides spp.*? Line 359: One space too many between "ratio" and "The" Line 366: "Figure 10"

Line 369: "Figure 10"

Line 398: "Figure 11"

Line 400: space

Line 401: "SW Indian Ocean"

Line 426: Bracket after Figure 6 missing Line 427: Bracket in the wrong place

*Thanks for pointing out the above errors.*

Line 430 - 432: rephrase sentence e.g.: "Although the leaning procedure by Barker et al. (2003) has been widely used (e.g., …) the removal of Mn-Mg coatings is still inefficient (Hasenfratz et al., …)."

*Thanks for the suggestion, we will change the wording.*

Line 439 – 441: rephrase sentence e.g.: "The high Fe/Ca ratios as well as the high Al/Ca ratios in most samples of all species used here (Table 2) indicate inefficient removal of silicate contaminants, suggesting that the number of rinse/ultrasonication repetitions of

the Barker et al. (2003) procedure is inadequate."

*Thanks for the suggestion, we will change the wording.*

Line 449: what is with the value of 0.15 mmol mol-1 in Table 2 for the lowest range of *Uvigerina peregrina*?

*Thanks for noticing this. This is a mistake; the range should be from 0.02 to 2.04 mmol mol-1 in Uvigerina peregrina.*

Line 450: add "(… 0.35 mmol mol-1; Table 2)"

Line 451: rephrase term: compared to Fe/Ca rations in *Cibicidoides wuellerstorfi* below 0.04 mmol mol-1.

Line 455/457: Do you mean Table A3?

*Yes, thanks for pointing this out. We will correct this.*

Line 457: add "(0.13 and 0.31 mmol mol-1)" behind *Cibicidoides spp.* -> There is also a dot missing after "*spp*"

Line 465: write: "… core depth, water depth, and morphology"

*Will be corrected.*

Line 474-476: Don't understand this sentence. What is a nearby region here?

*Thanks for highlighting this problem. We will rephrase the sentence.*

Line 477: I think there is a comma missing between contamination and Fe/Ca …?

*Yes, sentence will be corrected*

Line 478: Bracket closed after "Figure 8"

Line 479: Bracket closed after "Figure 7"

Line 498: Missing commas before and after "respectively", as well as before "but"
Line 507: one space too many between "i.e." and "*Cib*"

*Will be corrected.*

---

## Author Comment (AC2)

RC3

In "Western Indian Ocean bottom water temperature calibration – are benthic foraminifera Mg/Ca ratios a reliable palaeothermometry proxy?" Larsson & Jung measure Mg/Ca ratios in benthic foraminifers across a depth / space transect in the Western Indian Ocean to determine if a local Mg/Ca-temperature calibration is appropriate. They also test several cleaning techniques to determine their effects on contamination, sample yield, etc.

I'm no expert in Mg/Ca paleothermometry, but it seems like the half of the paper devoted to cleaning procedures is thorough and useful (if a little hard to follow). The calibration, however, is based on very few points (and anchored by one high-temperature sample), which the authors acknowledge, and is not likely to be used on its own. I think the exercise is still valuable, given that most of their data fall in the Mg/Ca-temperature space in prior calibrations, but the manuscript needs a clearer through-line, e.g., "best" cleaning protocol established -> despite efforts, majority of the specimens contaminated -> resulting calibration is sparse, but (most of) the data seem reasonable and species- / habitat-specific contamination thresholds, calibrations, etc. are recommended.

I believe that this can be of use to the paleoceanographic community pending major revisions in terms of structure, organization, and clarity – I would also highly recommend a thorough grammatical overhaul, there are numerous minor issues only some of which I've noted below. I don't believe further laboratory analyses are required, although there are one or two instances where they might be helpful.

*Thank you for your valuable input and for taking the time to thoroughly review our paper.*

*Your input on organisation and structure will be taken into account. We have aimed at making a balance between the methodological part of the paper and the results that as you have mention might still be of use despite contamination as some of the results agree with previous studies. However, with regard to the cleaning methods, we have not tested a wide range of approaches. We have followed one established cleaning procedure and adjusted the timings to accommodate for potentially more/less contamination in samples.*

General comments:

The title is rather vague and doesn't reveal much about the study's true findings. Maybe something like "Persistent contamination issues preclude a simple benthic Mg/Ca-temperature calibration in the Western Indian Ocean?" I'm sure you can come up with something better.

*Thanks for this useful suggestion. The next version will have an improved title.*

Your tests of the various cleaning procedure parameters are a major part of the paper, and I would mention it in the title. I think you need to emphasize that this is a valuable contribution to Mg/Ca thermometry, however – it reads to me like a sidenote compared to the calibration until you reach the later part of the manuscript.

If I missed this, I apologize, but any ideas as to why the other Indian Ocean calibrations didn't have as extensive of contamination issues?

*The tests of the various cleaning procedure parameters are indeed a major part of the paper however, as the methodology that is used is closely following the methodology developed by Barker et al. 2003, the main point of the paper is to highlight how the methodology by Barker et al. 2003 is adopted to*

*different samples and that more research is needed to explore what adaptations are needed for different samples, it also proves that calibrations comparison might be more difficult if different adaptations have been made in response to varying contamination.*

*Since no comments on adaptations to Barker et al's (2003) methodology have been mentioned we have assumed it has followed exactly the procedure of Barker et al., 2003. We don't know why previous papers have not run into these issues of contamination, as the cleaning methodology , especially applied to benthic foraminifera is generally known to be more difficult and suggested to require more rigorous cleaning. One possible reason is that specimens from the Indian Ocean calibrations have had significantly lower contamination prior to the cleaning methodology. Another explanation is if the foraminifera tests from the Indian Ocean have been deposited in sediments of lower silicate.*

Overall, the figures are well-made and easy to understand. I'm unconvinced this is a good idea, but if you (very lightly) shaded the "contaminated zone" above the peach-colored line e.g., in Figure 6, would it help drive home that almost everything's contaminated, or would it just add clutter?

*Adding a feature in the graphs to help convey the extent of contamination is a good idea.*

Line by line comments:

*We thank the reviewer for the very thorough assessment of our manuscript. I relation to the minor revisions comments, we will address all purely editorial issues raised in the list below and correct the text accordingly. Additional comments are added below.*

Lines 24-31: This paragraph seems out of place; reading it, I thought this paper was about to go in a very different direction. I think you could start from line 32 and be fine.

Lines 94-97: This whole paragraph or a statement of this kind belongs further towards the beginning – maybe at the end of Section 1?

Line 121 (?): What is the small inset panel on the right? I assume it's ship tracks but the information would be good to have in the caption.

*Thanks for this comment. The inset panel displays the positioning of the T and S profiles within the GLODAPv2 database. We will clarify this in the next version of the manuscript.*

Lines 153-216: Can you divide Experiments 1-3 into their own subheadings? I.e., "2.2.1. Preparation experiment 1: XYZ?" As it is now, it's a massive section that's difficult to follow. I could also see this being divided up where you explain the experiments simply and clearly in Section 2.2 and describe your findings in the Results.

*We are grateful for your comment and include it in the revised version of the manuscript.*

Line 219: There's inconsistency in foraminifera abbreviations: G. ruber vs. Cibicidoides wuellerstorfi, for example. I would go with the abbreviation, but be consistent either way.

Line 265: Where's panel D?

*Thanks for spotting this error. We will correct it.*

Line 325: Capitalize "c" in "c. wuellerstorfi."

Line 369: "Figure ???11"

Section 4.2. header: Not sure why this is blue?

Lines 467-472: Is there any support for this in the literature or is it speculation? I hate to ask, but any possibility of elemental mapping to support?

*Thanks for these comments. Elemental scanning is unfortunately beyond the scope of the project, but we will add references supporting the habit statements.*

Lines 478-479: Missing parentheses.

Lines 484-493: I think Section 4.8 belongs in the introduction – it's critical motivation for your cleaning tests but you don't bring it up until the end.

Lines 495-521: This is too long of a block of text, it's dense and hard to follow.

Lines 523-535: This is a great conclusion and summary! Bring some of this clarity to the introduction and method explanations.

*Thanks for this comment. It is much appreciated.*

---

## Author Comment (AC3)

RC2

EGUsphere – May 2024

***Western Indian Ocean bottom water temperature calibration – are benthic foraminifera Mg/Ca ratios a reliable paleothermometery proxy?***

**Larsson, V., Jung, S. Overall Review:**

This study presents some new Mg/Ca (and other elemental/Ca data) from benthic foram samples from core-top locations in the Western Indian Ocean basin and compares it to previously reported values from other locations in the wider Indian Ocean and beyond. They walk through three experimental methods for cleaning prior to Mg/Ca analysis and compare results in the context of contamination and data exclusionary procedures. They show agreement with some previously published calibrations for Mg/Ca – Bottom Water Temperature. This is a significant amount of work and the authors do have some exciting things to share.

However, their data is quite limited after removal of samples with apparent contamination, such that their final calibration model only includes 4 samples. As far as a method-based paper discussing the range of cleaning methods and implications of those methods on data collection, this paper is valuable. As far as a new model for reconstructing temperatures, this data should be taken with caution due to such low final sample numbers. However, their data do seem to fall within previously published values which shows good continuity. They could stress this more in the abstract as well, and make it very clear that their new calibration model contains only n = 4 samples.

I believe the paper should be published, but first with edits. Specifically, some of the text is confusing to follow as written, with many subsections in the discussion that are quite disparate/not well connected. If the authors are able, it would be good to go back through the discussion in particular to draw out and make very clear the major findings of their work. In general, a clearer outline of what this study brings to the literature/scientific field and the suggestions/findings for future work is needed. The Summary and Conclusions section is quite sparse in this regard and could be used as a place to discuss their findings in greater depth.

With respect to the methodological testing (cleaning methods), it would be helpful if the authors can work to make it far clearer what the difference between experiment 1 cleaning versus experiment 2

versus experiment 3 cleaning methods were – it's difficult the way it is worded now to follow. Perhaps a table outlining the differences of each procedure, and which steps were included in each so people can easily cross compare the methods would be instructive. Is the only difference that the methanol washes timing is reduced from 1-2 min to 20 seconds? Is it expected that the cleaning procedure would work the exact same way on the benthic species? Can the authors make a note of their expectations on this and reasoning for using *G. ruber* as their cleaning methodology species?

Please also update figures throughout to make it clearer for the reader to distinguish between datapoints, as they stand now many of the symbols are too small and soo

timilar to one another.

Overall, it is difficult to determine whether they conclude that benthic foraminifera Mg/Ca ratios are a reliable paleo thermometer in the Indian Ocean. Please be more explicit in your conclusions about your findings and how they fit within the larger context of benthic foram Mg/Ca paleothermometery.

*Thank you for the valuable input on our paper.*

*We agree that the discussion as well as the summary and conclusions section would benefit from a thorough rewording aimed at a clearer assessment and description of the main findings of this study.*

*A table outlining the difference in methodologies between experiment 1-3 will be considered. It is expected that the methodology used for planktic species works the same way for benthic species, however it might need more rigour (according to Barker et al., 2003). As discussed in the paper, several factors control the level of rigour needed (e.g morphology, species, pristine state of tests, sediments deposited in; these factors are directly and indirectly controlled by characteristics of benthic/planktic species). Because of this we have tested the methodology on benthic species in Experiment 3 and adjusted the methodology based on these test results. Previous studies e.g. Elderfield et al. (2010) and Healey et al. (2008) have followed the procedure of Barker et al., 2003 with no specifications of changes for the benthic species analysed.*

*The difference between Experiment 1-3 is the timing of methanol rinses, water rinses, ultrasonication, the treatment of shells prior to chemical cleaning, and acid cleaning Eppendorf tubes. We will ensure that these are better explained, either by adding table (see earlier comment) or a flow chart.*

**Detailed Review:**

**Figure edits:**

**Figure 1 –** maybe label with (a) and (b) so that it's clear beyond colour which map is showing which core- tops.

*Figure 1 – We agree, a) and b) labels will be added.*

**Figure 2 –** label with (a) and (b) (c) so it's clear in your figure caption when you are speaking about temperature and salinity and the transect map.

*Figure 2 – We agree, labels will be added*

**Figure 3 –** Be very clear in your figure caption. Indicate when you are referring to the 'blue arrow' and 'red arrow' because readers may think you are referring to the water temperature blue and red colours.

*Figure 3 – noted*

**Figure 4 –** Perhaps it might be easier to tell the difference between crushed versus not crushed samples if you colour them differently or have one filled in one empty triangles?

Also - does it really make sense to have one regression line that includes both crushed and un-crushed data points - what about two separate regressions showing the relationships across treatments?

Figure 4 caption: Is there a way to quickly describe the procedure here instead of referring to it as from Experiment 2? perhaps you can say using the method with shorter methanol cleaning step?

*Figure 4 – Thanks for this very useful input, will be edited accordingly. Filled and empty triangles would indeed clarify the messaging. We also assess how to quickly describe the procedure – if it is possible without too much text duplication.*

**Figure 6 -** In all species evaluated across all cores? indicate this if that's the case. Otherwise, where (which cores) are these data from?

*Figure 6 – Not all species are evaluated across all cores due to not all species found across all cores and size fraction, but we agree that we can include a table to provide information of which cores each sample is from.*

Also – panel D doesn't exist – where you indicate in your caption there should be the correlation between total Ca and contamination (Fe/Ca)

Rather than just saying 'horizontal lines' say "Orange lines show..." and indicate which value corresponds to which Element/Ca ratio.

*Panel d has been removed from a previous draft, this is an error in the caption and will be removed in edited version.*

*Reference to horizontal lines will be edited according to your advice.*

**Figure 8 –** The legend box is outside of the figures, try to align so it does not cross the figure border the way it does now.

*Figure 8 – noted.*

**Figure 9 –** How many samples make up each relationship – indicate sample number with " n = " Capitalize the C. for C. wuellerstorfi on figure

On line 329: should also include (u) here after Mg/Ca for (uncontaminated) indicator you have in your figure legend.

*Figure 9 – noted. Sample numbers will be added.*

**Figure 10 -** So all calibration models/ data presented are from this one species? make sure that is clear in the text above

Include reference/citation for the S. W Indian Ocean grey samples in your figure legend as you do for the purple and blue.

*Figure 10 – Yes this is referring to a single species. We will double check again but are unsure why this is unclear. C. wuellerstorfi is the only species discussed in the paragraph above and caption also only states C. wuellerstorfi. Both blue and grey are referring to Elderfield et al. 2006.*

**Line by line edits:**

*We thank the reviewer for the very thorough assessment of our manuscript. I relation to the minor revisions comments, we will address all purely editorial issues raised in the list below and correct the text accordingly. Additional comments are added below.*

**14:** the word entailed doesn't seem appropriate. Perhaps the word elucidated?
*Noted.*

**1G:** with what error can the 'wider' Indian ocean calibrations be used in the Western Indian Ocean? Could report that error here for brevity and maximal impact of abstract

*This would indeed improve the findings of the paper. We will investigate this. Thanks for the suggestion.*

**22:** With what error can BTW be reconstructed? What does your calibration error translate to in degrees C error of the BTW?

*We will address this in the final version of the manuscript.*

**24:** remove the word 'the' after controls…

**25:** the re-distribution of heat **in the oceans** is an ….

**28:** sentence should read: "… of the sensitivity and changes in thermohaline circulation. For example, on glacial-interglacial time scales, NADW and AABW…"

*Noted.*

**34:** can use $\delta^{18}O$ here instead of 'stable oxygen isotopes because you've already defined it earlier

*Noted.*

**41:** perhaps instead of "being developed" saying "is still unresolved" is better?

*Will be considered.*

**43:** Before starting in on the discussion of Mg incorporation, make clear that forams make calcite tests which is a calcium carbonate matrix… and describe that Mg substitutes for Ca in the lattice.

*Thanks for pointing this out*. We will add this to the final version.

**43:** remove the word "also" after "Mg2+ are also incorporated…"

**44:** indicate that Mg/Ca does not just depend on Mg/Ca of seawater and elemental partitioning…it's an endothermic process that relies on water temperature – hence the

reason we can use it to reconstruct water temperatures!

*We will clarify this.*

**48:** Define BWT here before using the acronym as it has not yet been defined in the text.

*It has been defined on line 40.*

**49:** Temperature appears to be the dominant environmental factor driving what? Indicate you mean the incorporation of Mg?

*Thanks for highlighting this, we will clarify it.*

**52:** carbonate ion saturation being dominant of/at what? What process is it affecting? Mg incorporation? Explain

*We will clarify this.*

**54:** What do you mean by 'spatially-varying' please elaborate here.

*Varying with depth and geographic location.*

**71:** "larger lowering" is awkward wording, consider rephrasing this

**80:** ".. determining **bulk or whole specimen** calcite Mg/Ca ratios"

**84:** "*Cibicidoides wuellerstorfi* **has** been one of..."

**85:** provide example citations for the use of benthic species for stable $\delta^{18}O$ and $\delta^{13}C$ reconstructions.

**87:** when referring to Mg/Ca incorporation – it is better to say Mg/Ca signatures OR Mg incorporation.

**91:** "... being usable..." is an awkward way of saying this. Perhaps "employed"

**92:** "Uncertainties remain (Stirpe et al. 2021)" – be clearer here, what uncertainties are you referring to that are raised in Stirpe et al. 2021.

**92:** "... entailing the need..." entailing is an awkward word choice: perhaps 'pointing to the need' or 'elucidating the need'

**94:** remove the word help before 'improving' and change 'improving' to 'improve'

**95/96:** you compare to calibrations from the Indian Ocean – are these previously published calibrations? If so list the citations if they are not too lengthy?

**108:** should say... "...core-top transect is comprised of..."

**108:** you only indicate 2 water masses in this first list... and go on to introduce the third later but it is slightly confusing/misleading to say three here and not list all three

*We will clarify this.*

**113:** where are the CTD temperature data from? When were they collected relative to the core-top collections?

*We will clarify this.*

**134:** no need for brackets around "where possible" when you are using commas

**144:** NIOP929 – what is this? Which core? From what cruise/expedition/researchers?

**145:** The sentence should read: "The samples were wet sieved over…"

**147:** The sentence should read: "….remove silicates, a hydrogen peroxide treatment to remove organic matter, and followed by an acid …"

**14G:** "depend" should be "depends"

**153:** The sentence should read: "…except for a reduction in duration of the methanol washes (25 seconds…"

**154:** The language "following previously analysed samples in the laboratory." Is not clear – please clarify and/or rephrase.

**159:** for how long was the hydrogen peroxide treatment? At what temperature?

*We will clarify this.*

**166:** Does this also mean you lose Mg in the glass slide method as well as Ca in order to keep the ratios similar to the pin method this must be true?

*We will clarify this.*

**178:** If the data in Figure 4 below show experiment 2 results, then pointing to it here for experiment 1 calcium values doesn't make sense

*We will correct this.*

**183:** Did you run a sensitivity test/ leverage test to see how much leverage that outlier has on your regression/model? if it's low then you can likely include it, if it's high then it's likely skewing the results.

*Thanks for this suggestion. It will be considered during the revision of manuscript.*

**184:** Is it really true that there is no relationship between Al/Ca and Mg/Ca? or is it just non-linear like logarithmic? Looks like it would be.

*We will re-assess and clarify in the next version of the manuscript.*

**185:** The correlation for not crushed is only really strong if the outlier is included, otherwise just clusters... the correlation for crushed tests is linear however

**224:** Please describe in more detail what the standards were - you should have that information from the lab in which these samples were run and it's standard practice to include the type/name of the specific standards used.

*Will be included in the next version.*

**225:** You should rename this Figure B1 - to indicate you are referring to Figure 1 of Appendix B and not Figure 1 of the manuscript.

**229:** 'effect' should be 'effects'

**Section headings 3.1 and 3.2** - you refer to more than just Mg/Ca ratios in these sections, perhaps re- naming to say elemental ratios or something similar?

**261-264:** It does not look like the samples are below the contamination thresholds? you even indicate in table 2 which samples are eliminated due to contamination     so this statement can't be true.

*We will clarify this.*

**298:** $r^2$ is a different metric from $R^2$ – you have used both throughout the text, so go through and ensure

**310:** Perhaps a sentence following indicating how many samples remained for the core-top calibration is useful here.

**318:** capitalize the 'f' in figure 9 – Figure 9

**323:** This calibration is based on 4 samples, one of which is quite far away from the others - could be contributing significant leverage to the calibration       i.e. without it the entire relationship would fall away. Have you tested this? Also, if you are going to report this calibration model (or any throughout the manuscript) it is good practice to indicate the number of samples/data points you have that built the model.   n this case: n = 4.

*We will address this in the next version of the manuscript.*

**348:** Maybe rather than 'abnormally' say anomalously high, not abnormally since you have cited evidence where similar reports have been made

**350-354:** So did you exclude this sample from calibration models? make it very clear if so.

**355- 360:** So did you exclude this sample from calibration models? make it very clear if so.

**362:** Are these studies also reporting only c. wuellerstorfi calibration models or which species are they reporting? indicate this here if they are species-specific or mixed-species models!

**369:** you have ??? after Figure in your reference to figure 11

**393:** the word maybe should be two words in this context: may be

**426:** ) close (Figure 6) with a bracket

**428:** under what subsection/subtitle number are you pointing readers to for discussion on degree of contamination?

*This will be clarified in the next version of the manuscript.*

**474:** This first sentence is awkward wording...do you mean to say your core top samples are closely located to previously published data?

*Will be reworded.*

**514:** risk should be risks

**517-518:** How? if you are to suggest more work should be done it's great if you can point to examples of next steps

**520:** benthic/planktic should be benthic versus planktic

**526-527:** Also include that this calibration only includes 4 samples.

**534-535:** It would be great to say something more about your study, while you are not able to speak on the other uncertainties/limiting factors, you have shown here that cleaning procedures may need to be refined and conducted species-by-species.

*Thanks for this suggestion, which we will take into account in the next version of the manuscript.*

---

## Author Response (AR1)

Review: "Western Indian Ocean bo6om water temperature calibra; on – are benthic foraminifera Mg/Ca ra;os a reliable palaeothermometry proxy?" Larsson & Jung, 2024 The manuscript provides a new dataset of various element/Ca ra;os, focussing on Mg/Ca ra-; os, for three benthic foraminifera species. The data originate from the western Indian Ocean and were checked by the authors for their applicability as a paleothermometer proxy for bottom water temperatures. Although the number of samples seems too small to establish a valid new calibra;on, the manuscript contributes to an improved understanding of benthic foraminifera and their usability as palaeoproxy in this area. Furthermore, the authors compare their data set with two others from the region and evaluate the quality and relevance of their data as a palaeoproxy in the discussion chapter, under considera; on of significant literature. As I myself am not par; cularly familiar with benthic foraminifera, I cannot make a qualita; ve statement about the methodology which is used here. The authors have presented their results in a sufficient number of graphics, although some of them s;ll need a bit of reworking. Furthermore, the language is a li6le clumsy in some places, but as I am not a na; ve speaker myself, I have only made minor comments here. Overall, I rate the manuscript as good and recommend publica; on, although there are s;ll a few points that need to be improved.

2 Be more consistent throughout the manuscript e.g.: 1) Line 172 "(Table A1 in Appendix A)" vs. Line 201 "(Appendix A Table A1)" 2) Abbrevia; on of the foraminifera. Some; mes you wrote Cibicidoides mundulus vs C. mundulus. You can write the en; re name throughout the manuscript, but I recommend to write the full genus and species name when you first men; on it and then con; nue throughout the manuscript with the abbrevia; on

3) Same for figure vs. fig. vs Figure (and table, Table, tab.), choose one and stay consistent. Figures and Tables Figure 2 would benefit from gridding the poten; al temperature. Figure 4: Use either two different symbols or two different colors (best is both), it is hard to dis; nguish between both. Figure 6: Same like Figure 4. I find it hard to dis; nguish between the both triangles, since the dark blue and black do not show a strong contrast, but also the other blue symbols could benefit from a different color. Also, the graphic labelling is confusing here. I would write "Mg/Ca ra;o against (a) Fe/Ca ra;os, (b) Al/Ca ra;o and (c) Mn/Ca ra;o in Cibicidoides spp. ..." Figure 9: Again. Maybe use a circle for C. spp. ? Table 1: "Wuellerstrofi" is wri6en in capital le6ers, change it. Table A2: Numbers e.g. at 2a, 2g and 3a seem to be smaller. Table A2: Why is the species at samples from 3g to 3i G. ruber? I thought it is Uvigerina spp. and Cibicidoides spp. 3 Minor revisions: Line 74: One space too many ader "i.e." Line 92: strange bracket ader "(Stripe et al., 2021)" and one space too many Line 107-109: You say "three" water masses but listed just "two" of them Line 136: two dots at the end of the sentence Line 138: One space too many ader "(Table 1)" Line 143: first men; on of G. ruber -> write the en;re name Line 153: write "In experiment 1" to keep it consistent. Furthermore, men; on that you used 6 sets with a varying amount of G. ruber somewhere. Line 162-164: 5.81 ppm and 9.53 ppm, where does these numbers come from? In Table A2 I see three tests with G. ruber for the crushing experiment between two glass slides (Ca 0.55, 7.5 and 3.75 ppm = 3.93) and three tests with G. ruber for the crushing experiment using a metal pin (1.68, 5.1, 5.66 = 4.15). Same for Mg/Ca, can't find the 3.43 and 3.53 mmol mol-1 in A2, especially since sample 1a has incredibly high values of 35.23 mmol mol-1. Line 170: maybe add "(0.55 to 7.50 ppm in both crushing between two glass slides

**Commented [VL1]:** Not sure what is the best format to use. I think U. spp is a bit too short. also C. spp. Check with Simon

and when using a metal pin, Table 1A and A2 in Appendix A)" to make sure that the general results, regardless of the method used, are really low. Line 180-184: For your average values 0.38, 2.91 and 3.01 mmol mol-1, I have 0.37, 2.90 and 3.02 respec; vely. I suppose this is because you used a higher number of decimals in your original calcula; on. I am men; on this in case you want to change it, but I think this is fine. For your mean Fe/Ca ra; o of 0.61 mmol mol-1, I have 0.57 mmol mol-1 instead. Please check this again and correct it.

Line 199: (6x25): change sentence: "In the procedure, specimens of Globigerinoides ruber (6 sets with a varying amount of 10 - 50 individuals; Table A1, Appendix) picked from ..."

Line 200: remove point ader "Uvigerina..."

Line 201: (Table A1 in Appendix A) -> see Line 172

Line 214: Make it two sentences: "... proposed by Hasenfratz et al. (2017). This suggest that Mn-oxide..."

Line 228 230: change to (Figure B1 in Appendix B)

Line 262: One space too many

Line 280: Table A3?

Line 299-300: I see four samples in water depth deeper than 2500 m and five samples in water depth 2500  $\rm m$  ..."

Line 303: Cibicidoides in italics

Line 308 – 310: rephrase sentence: "The Mg/Ca ra; os of Cibicidoides spp., although higher in their Fe/Ca ra; os than >0.1 mmol mol-1, were also included, since they show no correla; on between Mg/Ca ra; os and Fe/Ca ra; os."

Line 310: One space too many between "Table" and "2". Also, write "Figure 6" in capital le6ers to stay consistent.

Line 312: C. mundulus and C. wuellerstorfi, stay consistent.

Line 313: "Figure 9". Also, remove point ader "Cibicidoides". Rich;g: Cibicidoides spp.

Line 318: "Figure 9" Line 314 & 320: BWT Line 321: Sentence in parentheses not in italics

Line 322: "Figure 9" Line 331: Cibicidoides spp.

Line 337: Cibicidoides spp.; Also, one space too many between "Table 2" and "When"

Line 353: One space too many between "Table 2" and "It" Line 356: compared to what? Samples from Cibicidoides spp.?

Line 359: One space too many between "ra;o" and "The"

Line 366: "Figure 10" Line 369: "Figure 10" 5 Line 398: "Figure 11" Line 400: space

Line 401: "SW Indian Ocean" Line 426: Bracket ader Figure 6 missing

Line 427: Bracket in the wrong place

Line 430 - 432: rephrase sentence e.g.: "Although the leaning procedure by Barker et al. (2003) has been widely used (e.g., ...) the removal of Mn-Mg coa;ngs is s;ll inefficient (Hasenfratz et al., ...)."

Line 439 – 441: rephrase sentence e.g.: "The high Fe/Ca ra; os as well as the high Al/Ca ra; os in most samples of all species used here (Table 2) indicate inefficient removal of silicate contaminants, sugges; ng that the number of rinse/ultrasonica; on repe;; ons of the Barker et al. (2003) procedure is inadequate."

Line 449: what is with the value of 0.15 mmol mol-1 in Table 2 for the lowest range of Uvigerina peregrina? Line 450: add " $(\dots 0.35 \text{ mmol mol-1}; \text{Table 2})$ "

Commented [VL2]: kept old averages because correct. But change Fe/Ca to 0.57, mistake in calculation

Commented [VL3]: Changed. Also sentence on line 195 clarified

Commented [VL4]: done

Commented [VL5]: done

Commented [VL6]: done

Commented [VL7]: Not changed, should be Table as is written

Commented [VL8]: now table A4

Commented [VL9]: Checked and changed

Line 451: rephrase term: compared to Fe/Ca ra; ons in Cibicidoides wuellerstorfi below 0.04 mmol mol-1. Commented [VL10]: fixed Line 455/457: Do you mean Table A3? Commented [VL11]: fixed Line 457: add "(0.13 and 0.31 mmol mol-1)" behind Cibicidoides spp. -> There is also a dot missing ader "spp" Commented [VL12]: added Line 465: write: "... core depth, water depth, and morphology" Commented [VL13]: added Line 474-476: Don't understand this sentence. What is a nearby region here? Commented [VL14]: clarified Line 477: I think there is a comma missing between contamina; on and Fe/Ca ...? Line Commented [VL15]: added 478: Bracket closed ader "Figure 8" Commented [VL16]: added Line 479: Bracket closed ader "Figure 7" Commented [VL17]: added Line 498: Missing commas before and ader "respec;vely", as well as before "but" Commented [VL18]: added Line 507: one space too many between "i.e." and "Cib" Commented [VL19]: fixed

**Referee2**

14: the word entailed doesn't seem appropriate. Perhaps the word elucidated? -Rephrased sentence

19: with what error can the 'wider' Indian ocean calibrations be used in the Western Indian Ocean? Could report that error here for brevity and maximal impact of abstract No idea – ask Simon

22: With what error can BTW be reconstructed? What does your calibration error translate to in degrees C error of the BTW?

- Ask Simon how to appropriately make conversion

24

Fixed

25

Fixed

28

Fixed

34

- Fixed

41

Fixed

43

Fixed

- Fixed

44

Sentence clarified as suggested, but flow can be improved if there is time.

| 48 -                                                | Defined                                                                                                                       |
|-----------------------------------------------------|-------------------------------------------------------------------------------------------------------------------------------|
| 49
-                                             | Fixed                                                                                                                         |
| 54
-                                             | Fixed                                                                                                                         |
| 71
-                                             | Rephrased sentences                                                                                                           |
| 80
-
84                                       | Fixed                                                                                                                         |
| -                                                   | Fixed                                                                                                                         |
| 85
-                                             | provide example citations for the use of benthic species for stable $\delta {\rm 18O}$ and $\delta {\rm 13C}$ reconstructions |
| 87
-                                             | fixed                                                                                                                         |
| 91 -                                                | removed sentence as doesn't add much                                                                                          |
| 92
-
-
95-96
-fixed                     | rephrased sentences fixed                                                                                                     |
| 113
-fixed                                       |                                                                                                                               |
| 134
-fixed                                       |                                                                                                                               |
| 144
Fixed
145
Fixed                        |                                                                                                                               |
| 159 (162) How long water bath and what temperature? |                                                                                                                               |

224: Please describe in more detail what the standards were, I don't think I have this info - can I ask Laetitia? 225 done Section headings 3.1 and 3.2 --fixed 261-264: It does not look like the samples are below the contamination thresholds? you even indicate in table 2 which samples are eliminated due to contamination.... so this statement can't be true. -yes this sentence is true. This is refering to Cibicidoides spp. samples that contained specimens mixed Cibicidoides species 298 -It should be r2 310 -go back to again – write a sentence and clarify how many samples were left included in the calibration at the end. How many included in cibicidoides wuellerstorfi? How many samples were left if following the contamination thresholds? done 323 -go back to again: does the calibration fall away if I remove the datapoint at high temperature? (likely yes) Add a sentence stating this? Ask Simon -go back to again; add sentences clarifying which are excluded/included. 362 -fixed 369 -fixed 393 -fixed 426 -fixed

428 -added 474

-need to check in odv

514

-fixed

**Referee 3**

Line by line comments:

Lines 24-31: This paragraph seems out of place; reading it, I thought this paper was about to go in a very different direction. I think you could start from line 32 and be fine.

-Have rearranged and changed first paragraph

Lines 94-97: This whole paragraph or a statement of this kind belongs further towards the beginning – maybe at the end of Section 1?

-edited

Line 121 (?): What is the small inset panel on the right? I assume it's ship tracks but the information would be good to have in the caption.

-added

Lines 153-216: Can you divide Experiments 1-3 into their own subheadings? I.e., "2.2.1. Preparation experiment 1: XYZ?" As it is now, it's a massive section that's difficult to follow. I could also see this being divided up where you explain the experiments simply and clearly in Section 2.2 and describe your findings in the Results.

-added Table 2 clarifying differences

Line 219: There's inconsistency in foraminifera abbreviations: G. ruber vs. Cibicidoides wuellerstorfi, for example. I would go with the abbreviation, but be consistent either way.

-fixed

Line 265: Where's panel D?

-removed

Line 325: Capitalize "c" in "c. wuellerstorfi."

-fixed

Line 369: "Figure ???11"

-fixed

Section 4.2. header: Not sure why this is blue?

-fixed

Lines 467-472: Is there any support for this in the literature or is it speculation? I hate to ask, but any possibility of elemental mapping to support?

Thanks for these comments. Elemental scanning is unfortunately beyond the scope of the

project, but we will add references supporting the habit statements.

Lines 478-479: Missing parentheses.

-fixed

Lines 484-493: I think Section 4.8 belongs in the introduction – it's critical motivation for your cleaning tests but you don't bring it up until the end.

Lines 495-521: This is too long of a block of text, it's dense and hard to follow.

-rephrased

Lines 523-535: This is a great conclusion and summary! Bring some of this clarity to the introduction and method explanations.

-clarified intro

---

## Referee Report (RR1)

**Review:**

"Persistent contamination in benthic foraminifera-based thermometry using standard cleaning methods"

Larsson and Jung, 2025

In the manuscript, the authors dealt with the preparation of benthic foraminifera for Mg/Ca thermometry. Therefore, they picked benthic foraminifera of the genus *Uvigerina* and *Cibicidoides* from sediment core top samples from the western Indian Ocean and measured the Mg/Ca ratio after different cleaning procedures.

The manuscript is well structured and organized. In my opinion, although the authors used a rather small data set, they did a good job comparing the main cleaning steps using the most important references of Mg/Ca cleaning. I also have the impression that the style and language (I am not a native speaker myself) have improved significantly compared to the previous manuscript (which I also reviewed).

All in all, I found only a few things to correct and approve a publication after a small revision.

**Minor Revision:**

Line 46: 106 und 107 -> 10 Ma und 100 Ma

Line 65: The oxidative cleaning is usually used for the removal of organic matter (see Barker 2003) -> change sentence: "... and an oxidative cleaning step to remove organic matter"

Line 85: Reference to Graphic 1 feels a bit out of place as you are only talking about species-specific calibrations, but these are not highlighted in the graphic. I would refer to graphic 1 later.

Line 99: Figure 1 Write down the name NIOP in the map, so that these cores are easier to find.

Line 108: "... Upper CDW (UCDW)"

Line 137: "s" is missing in specimens

Line 138: Correct bracket content (i.e., 250-355 μm and 250-355 μm???)

Line 144-148: Rephrase e.g., "Due to a lack of sufficient amount of planktic and benthic foraminifera in our transect, we test the cleaning procedure on individuals of the planktic foraminifera species *Globigerinoides ruber* (*G. ruber*, 250-355  $\mu$ m) at core NIOP929 (Saher et al., 2009), located further north at the continental slope off Somalia (van Hinte et al., 1995; cite cruise report here)."

Line 258 and 263: stay consistent throughout the text. Remove "see" from (Table xy)

Line 262: in Table 3 stands 68.73

Line 262: Stay consistent. Either you write the exact number of the table (3.99) or you round up/down. 3.99 is going to 4.0 and 68.73 is going to 68.7.

---

## Author Response (AR2)

Response to comments made by all three referees. Comments are shown in Italics and are highlighted in green.

**RC1:**

**Review:**

"Western Indian Ocean bottom water temperature calibration – are benthic foraminifera Mg/Ca ratios a reliable palaeothermometry proxy?" Larsson & Jung, 2024

The manuscript provides a new dataset of various element/Ca ratios, focusing on Mg/Ca ratios, for three benthic foraminifera species. The data originate from the western Indian Ocean and were checked by the authors for their applicability as a paleothermometer proxy for bottom water temperatures. Although the number of samples seems too small to establish a valid new calibration, the manuscript contributes to an improved understanding of benthic foraminifera and their usability as palaeoproxy in this area.

Furthermore, the authors compare their data set with two others from the region and evaluate the quality and relevance of their data as a palaeoproxy in the discussion chapter, under consideration of significant literature. As I myself am not particularly familiar with benthic foraminifera, I cannot make a qualitative statement about the methodology which is used here.

The authors have presented their results in a sufficient number of graphics, although some of them still need a bit of reworking. Furthermore, the language is a little clumsy in some places, but as I am not a native speaker myself, I have only made minor comments here.

Overall, I rate the manuscript as good and recommend publication, although there are still a few points that need to be improved.

Thank you for carefully reviewing our paper and highlighting the inconsistencies.

**Be more consistent throughout the manuscript e.g.:**

1) Line 172 "(Table A1 in Appendix A)" vs. Line 201 "(Appendix A Table A1)"

**The referencing of Figures or tables in the Appendix has been amended according to the referee's suggestion.**

- 2) Abbreviation of the foraminifera. Sometimes you wrote *Cibicidoides mundulus* vs *C. mundulus*. You can write the entire name throughout the manuscript, but I recommend to write the full genus and species name when you first mention it and then continue throughout the manuscript with the abbreviation.
- 3) Same for figure vs. fig. vs Figure (and table, Table, tab.), choose one and stay consistent.

We thank the reviewer for both comments. Species names are now being referred to in a consistent fashion.

**Figures and Tables**

Figure 2 would benefit from gridding the potential temperature.

**We appreciate this comment and have now redesigned figure 2 according to the suggestion.**

Figure 4: Use either two different symbols or two different colors (best is both), it is hard to distinguish between both.

Figure 4 has been re-designed based on the suggestion by the referee.

Figure 6: Same like Figure 4. I find it hard to distinguish between the both triangles, since the dark blue and black do not show a strong contrast, but also the other blue symbols could benefit from a different color. Also, the graphic labelling is confusing here. I would write "Mg/Ca ratio against (a) Fe/Ca ratios, (b) Al/Ca ratio and (c) Mn/Ca ratio in *Cibicidoides spp.* ..."

Figure 4 has been re-designed based on the suggestion by the referee.

Figure 8 has been re-designed based on the suggestion by the referee.

Figure 9: Again. Maybe use a circle for *C. spp*. ?

Figure 9 — We have redesigned figure 9, which now includes color coding which is consistent with other figures as much as possible.

Table 1: "Wuellerstrofi" is written in capital letters, change it.

The error has been corrected

Table A2: Numbers e.g. at 2a, 2g and 3a seem to be smaller.

The error has been corrected.

Table A2: Why is the species at samples from 3g to 3i *G. ruber*? I thought it is *Uvigerina spp.* and *Cibicidoides spp.*

The error has been corrected.

**Minor revisions:**

We thank the reviewer for the very thorough assessment of our manuscript. In relation to the minor revisions comments, we will address all purely editorial issues raised in the list below and correct the text accordingly. Additional comments are added below.

Line 74: One space too many after "i.e." check again.

This section has been reworded and the error removed.

Line 92: strange bracket after "(Stripe et al., 2021)" and one space too many

This section has been reworded and the error removed.

Line 107-109: You say "three" water masses but listed just "two" of them

This section has been reworded and the error removed.

Line 136: two dots at the end of the sentence

The error has been corrected.

Line 138: One space too many after "(Table 1)"

The error has been corrected.

Line 143: first mention of *G. ruber* -> write the entire name

The error has been corrected.

Line 153: write "In experiment 1" to keep it consistent. Furthermore, mention that you used 6 sets with a varying amount of *G. ruber* somewhere.

This section has been reworded and the error removed.

Line 162-164: 5.81 ppm and 9.53 ppm, where does these numbers come from? In Table A2 I see three tests with G. ruber for the crushing experiment between two glass slides (Ca 0.55, 7.5 and 3.75 ppm = **3.93**) and three tests with G. ruber for the crushing experiment using a metal pin (1.68, 5.1, 5.66 = **4.15**). Same for Mg/Ca, can't find the 3.43 and 3.53 mmol mol-1 in A2, especially since sample 1a has incredibly high values of 35.23 mmol mol-1.

The Ca concentration data are based on the normalized data in table A2. This has been clarified in the text. The Mg/Ca averages have been corrected. We also added a passage stating that we regard sample 1a as an outlier. In relation to the quoted Mg/Ca data there was an error in the manuscript that has now been corrected. The respective passage has also been reworded to clarify the message.

Line 170: maybe add "(0.55 to 7.50 ppm in both crushing between two glass slides and when using a metal pin, Table 1A and A2 in Appendix A)" to make sure that the general results, regardless of the method used, are really low.

Thanks for this suggestion. We have changed the text accordingly.

Line 180-184: For your average values 0.38, 2.91 and 3.01 mmol mol-1, I have 0.37, 2.90 and 3.02 respectively. I suppose this is because you used a higher number of decimals in your original calculation. I am mention this in case you want to change it, but I think this is fine. For your mean Fe/Ca ratio of 0.61 mmol mol-1, I have 0.57 mmol mol-1 instead. Please check this again and correct it.

Thanks for this comment. We have re-calculated the averages and corrected small rounding errors when needed.

Line 199: (6x25): change sentence: "In the procedure, specimens of *Globigerinoides ruber* (6 sets with a varying amount of 10 - 50 individuals; Table A1, Appendix) picked from ..."

**Thanks for this suggestion. We have changed the text accordingly.**

Line 200: remove point after "Uvigerina..."

We have changed the text accordingly.

Line 201: (Table A1 in Appendix A) -> see Line 172

We have changed the text accordingly.

Line 214: Make it two sentences: "... proposed by Hasenfratz et al. (2017). This suggest that Mn-oxide..."

We have changed the text accordingly.

Line 228 230: change to (Figure B1 in Appendix B)

We have changed the text accordingly.

Line 262: One space too many

We have changed the text accordingly.

Line 280: Table A3?

We have corrected the table referencing issues.

Line 299-300: I see **four** samples in water depth deeper than 2500 m and **five** samples in water depth <1500 m. Furthermore, instead of "Below 2500 m" maybe write "In water depth >2500 m ..."

We have changed the text accordingly.

Line 303: Cibicidoides in italics

We have changed the text accordingly.

Line 308 – 310: rephrase sentence: "The Mg/Ca ratios of *Cibicidoides spp.*, although higher in their Fe/Ca ratios than >0.1 mmol mol-1, were also included, since they show no correlation between Mg/Ca ratios and Fe/Ca ratios."

We have changed the text accordingly.

Line 310: One space too many between "Table" and "2". Also, write "Figure 6" in capital letters to stay consistent.

**We have changed the text accordingly.**

Line 312: C. mundulus and C. wuellerstorfi, stay consistent.

This has been corrected. Please see earlier comment on consistent species quotation.

Line 313: "Figure 9". Also, remove point after "Cibicidoides". Richtig: Cibicidoides spp.

We have changed the text accordingly

Line 318: "Figure 9"

Done

Line 314 & 320: BWT

We have changed the text accordingly

Line 321: Sentence in parentheses not in italics

This part has been reworded and the error removed.

Line 322: "Figure 9"

According to our understanding of the EGU guidelines, "In Fig. 9" should be correct.

Line 331: Cibicidoides spp.

This has been corrected. Please see earlier comment on consistent species quotation.

Line 337: Cibicidoides spp.; Also, one space too many between "Table 2" and "When"

This has been corrected. Please see earlier comment on consistent species quotation. The extra space has been removed.

Line 353: One space too many between "Table 2" and "It"

The extra space has been removed.

Line 356: compared to what? Samples from Cibicidoides spp.?

The section has been reworded.

Line 359: One space too many between "ratio" and "The"

The section has been reworded.

Line 366: "Figure 10"

According to our understanding of the EGU guidelines, "In Fig. X" should be correct.

Line 369: "Figure 10"

According to our understanding of the EGU guidelines, "In Fig. X" should be correct.

Line 398: "Figure 11"

According to our understanding of the EGU guidelines, "In Fig. X" should be correct.

Line 400: space

Space is removed.

Line 401: "SW Indian Ocean"

Done

Line 426: Bracket after Figure 6 missing

Done

Line 427: Bracket in the wrong place

**Corrected.**

Line 430 - 432: rephrase sentence e.g.: "Although the leaning procedure by Barker et al. (2003) has been widely used (e.g., ...) the removal of Mn-Mg coatings is still inefficient (Hasenfratz et al., ...)."

We have changed the text accordingly

Line 439 – 441: rephrase sentence e.g.: "The high Fe/Ca ratios as well as the high Al/Ca ratios in most samples of all species used here (Table 2) indicate inefficient removal of silicate contaminants, suggesting that the number of rinse/ultrasonication repetitions of the Barker et al. (2003) procedure is inadequate."

We have changed the text accordingly

Line 449: what is with the value of 0.15 mmol mol-1 in Table 2 for the lowest range of *Uvigerina* peregrina?

Thanks for noticing this. This is a mistake; the range should be from 0.02 to 2.04 mmol mol-1 in Uvigerina peregrina. This is now corrected.

Line 450: add "(... 0.35 mmol mol-1; Table 2)"

Done

Line 451: rephrase term: compared to Fe/Ca rations in *Cibicidoides wuellerstorfi* below 0.04 mmol mol-1.

**Done**

Line 455/457: Do you mean Table A3?

**Corrected.**

Line 457: add "(0.13 and 0.31 mmol mol-1)" behind *Cibicidoides spp. ->* There is also a dot missing after "*spp*"

**Corrected.**

Line 465: write: "... core depth, water depth, and morphology"

**Corrected.**

Line 474-476: Don't understand this sentence. What is a nearby region here?

**This section has been reworded.**

Line 477: I think there is a comma missing between contamination and Fe/Ca ...?

**Corrected.**

Line 478: Bracket closed after "Figure 8"

**Corrected.**

Line 479: Bracket closed after "Figure 7"

**We corrected the figure reference.**

Line 498: Missing commas before and after "respectively", as well as before "but"

**Corrected.**

Line 507: one space too many between "i.e." and "Cib"

**Corrected.**

RC2

EGUsphere – May 2024

Western Indian Ocean bottom water temperature calibration – are benthic foraminifera Mg/Ca ratios a reliable paleothermometery proxy?

**Larsson, V., Jung, S. Overall Review:**

This study presents some new Mg/Ca (and other elemental/Ca data) from benthic foram samples from core-top locations in the Western Indian Ocean basin and compares it to previously reported values from other locations in the wider Indian Ocean and beyond. They walk through three experimental methods for cleaning prior to Mg/Ca analysis and compare results in the context of contamination and data exclusionary procedures. They show agreement with some previously published calibrations for Mg/Ca – Bottom Water Temperature. This is a significant amount of work and the authors do have some exciting things to share.

However, their data is quite limited after removal of samples with apparent contamination, such that their final calibration model only includes 4 samples. As far as a method-based paper discussing the range of cleaning methods and implications of those methods on data collection, this paper is valuable. As far as a new model for reconstructing temperatures, this data should be taken with caution due to such low final sample numbers. However, their data do seem to fall within previously published values which shows good continuity. They could stress this more in the abstract as well, and make it very clear that their new calibration model contains only n = 4 samples.

**We have reworded the abstract, following the suggestions by referee 2**

I believe the paper should be published, but first with edits. Specifically, some of the text is confusing to follow as written, with many subsections in the discussion that are quite disparate/not well connected. If the authors are able, it would be good to go back through the discussion in particular to draw out and make very clear the major findings of their work. In general, a clearer outline of what this study brings to the literature/scientific field and the suggestions/findings for future work is needed. The Summary and Conclusions section is quite sparse in this regard and could be used as a place to discuss their findings in greater depth.

We have reworded large parts of the manuscript aimed at addressing the raised concerns.

With respect to the methodological testing (cleaning methods), it would be helpful if the authors can work to make it far clearer what the difference between experiment 1 cleaning versus experiment 2.

In order to improve clarity of the text in relation to used methodologies, we have added table 2, which summarizes the procedural steps involved in the different experiments. We have also reworded the text to improve clarity of messaging.

versus experiment 3 cleaning methods were – it's difficult the way it is worded now to follow. Perhaps a table outlining the differences of each procedure, and which steps were included in each so people can easily cross compare the methods would be instructive. Is the only difference that the methanol washes timing is reduced from 1-2 min to 20 seconds? Is it expected that the cleaning procedure would work the exact same way on the benthic species? Can the authors make a note of their expectations on this

and reasoning for using G. ruber as their cleaning methodology species?

Please see previous comment in relation to methodology. In relation to the use of G. ruber, we do state that this is the only species available in sufficient numbers to carry out testing of different cleaning procedures. There were simply not enough benthic foraminifera of any kind in the Tanzania samples to carry out these tests. We have also indicated the differences between the G. ruber and the various benthic foraminifera used.

Please also update figures throughout to make it clearer for the reader to distinguish between datapoints, as they stand now many of the symbols are too small and so similar to one another.

Please see earlier responses to similar requests made by referee 1. The figures have been updated to improve accessibility.

Overall, it is difficult to determine whether they conclude that benthic foraminifera Mg/Ca ratios are a reliable paleo thermometer in the Indian Ocean. Please be more explicit in your conclusions about your findings and how they fit within the larger context of benthic foram Mg/Ca paleothermometery.

We have substantially reworded the manuscript, hoping that this has improved the clarity of our messaging.

**Detailed Review:**

**Figure edits:**

**Figure 1** – maybe label with (a) and (b) so that it's clear beyond colour which map is showing which core- tops.

**Corrected.**

**Figure 2** – label with (a) and (b) (c) so it's clear in your figure caption when you are speaking about temperature and salinity and the transect map.

**Done.**

**Figure 3** – Be very clear in your figure caption. Indicate when you are referring to the 'blue arrow' and 'red arrow' because readers may think you are referring to the water temperature blue and red colours.

**This has been clarified.**

**Figure 4** – Perhaps it might be easier to tell the difference between crushed versus not crushed samples if you colour them differently or have one filled in one empty triangles?

**This has been addressed as part of the figure update.**

Also - does it really make sense to have one regression line that includes both crushed and uncrushed data points - what about two separate regressions showing the relationships across

**treatments?**

**We have added regression lines in accordance with this suggestion.**

Figure 4 caption: Is there a way to quickly describe the procedure here instead of referring to it as from Experiment 2? perhaps you can say using the method with shorter methanol cleaning step?

This has been addressed as part of the figure update. We have also indicated procedural differences, whilst avoiding too much repetition.

**Figure 6** - In all species evaluated across all cores? indicate this if that's the case. Otherwise, where (which cores) are these data from?

Not all species are evaluated across all cores due to not all species found across all cores and size fractions. The updated color coding should improve accessibility of the figure. Combined, tables 1 and 3 contain information regarding the origin of the samples as well the measurement data for each sample.

Also – panel D doesn't exist – where you indicate in your caption there should be the correlation between total Ca and contamination (Fe/Ca)

Rather than just saying 'horizontal lines' say "Orange lines show..." and indicate which value corresponds to which Element/Ca ratio.

Panel d has been removed from a previous draft, this was an error in the caption and has now been removed that. The description of the figure has been updated based on the referee's suggestion.

**Figure 8** – The legend box is outside of the figures, try to align so it does not cross the figure border the way it does now.

**Corrected.**

Figure 9 – How many samples make up each relationship – indicate sample number with "n = Capitalize the C. for C. wuellerstorfi on figure

On line 329: should also include (u) here after Mg/Ca for (uncontaminated) indicator you have in your figure legend.

We have corrected the figure and amended the figure caption including a providing a n-value for the C. wuellerstorfi based Mg/Ca-BWT relationship.

**Figure 10** - So all calibration models/ data presented are from this one species? make sure that is clear in the text above

Include reference/citation for the S. W Indian Ocean grey samples in your figure legend as you do for the purple and blue.

We have amended the figure caption more explicitly stating that this figure only shows C. wuellerstorfi related data.

**Line by line edits:**

We thank the reviewer for the very thorough assessment of our manuscript. Please find our responses below.

**14:** the word entailed doesn't seem appropriate. Perhaps the word elucidated? *The abstract has been reworded which removed the issue.*

**19:** with what error can the 'wider' Indian ocean calibrations be used in the Western Indian Ocean? Could report that error here for brevity and maximal impact of abstract

**22:** With what error can BTW be reconstructed? What does your calibration error translate to in degrees C error of the BTW?

Response to the last two comments. We agree with the reviewer that more statistical information will benefit the manuscript. The main message of the manuscript is that standard cleaning methods may not sufficiently remove contaminants. Based on our assessment of data, only four C. wuellerstorfi data points determine our tentative Mg/Ca calibration. Given the small "n" we want to avoid over emphasizing our calibration. We did, however, add the error in BTW's in the main body of text (when the calibration is introduced).

24: remove the word 'the' after controls...

This section has been removed.

25: the re-distribution of heat in the oceans is an ....

This section has been removed.

**28:** sentence should read: "... of the sensitivity and changes in thermohaline circulation. For example, on glacial-interglacial time scales, NADW and AABW..."

This section has been removed.

**34:** can use  $\delta^{18}$ O here instead of 'stable oxygen isotopes because you've already defined it earlier

In the updated section the  $\delta^{18}$ O needs to be defined here because the previous paragraph has been removed.

**41:** perhaps instead of "being developed" saying "is still unresolved" is better?

The section has been reworded which removed the problematic section.

**43:** Before starting in on the discussion of Mg incorporation, make clear that forams make calcite tests which is a calcium carbonate matrix... and describe that Mg substitutes for Ca in the lattice.

**Done.**

43: remove the word "also" after "Mg2+ are also incorporated..."

**Done.**

**44:** indicate that Mg/Ca does not just depend on Mg/Ca of seawater and elemental partitioning...it's an endothermic process that relies on water temperature – hence the reason we can use it to reconstruct water temperatures!

**This has been clarified.**

**48:** Define BWT here before using the acronym as it has not yet been defined in the text.

It is being defined in the preceding paragraph.

**49:** Temperature appears to be the dominant environmental factor driving what? Indicate you mean the incorporation of Mg?

This has been clarified.

**52:** carbonate ion saturation being dominant of/at what? What process is it affecting? Mg incorporation? Explain

This has been clarified.

**54:** What do you mean by 'spatially-varying' please elaborate here.

This has been clarified.

**71:** "larger lowering" is awkward wording, consider rephrasing this

This has been reworded.

**80:** ".. determining **bulk or whole specimen** calcite Mg/Ca ratios"

Suggestion is included.

**84:** "Cibicidoides wuellerstorfi has been one of..."

Has been corrected.

**85:** provide example citations for the use of benthic species for stable  $\delta^{18}{\rm O}$  and  $\delta^{13}{\rm C}$  reconstructions.

**References have been added.**

**87:** when referring to Mg/Ca incorporation – it is better to say Mg/Ca signatures OR Mg incorporation.

**Has been corrected.**

91: "... being usable..." is an awkward way of saying this. Perhaps "employed"

The section has been reworded which removed the problematic section.

**92:** "Uncertainties remain (Stirpe et al. 2021)" – be clearer here, what uncertainties are you referring to that are raised in Stirpe et al. 2021.

The section has been reworded which removed the problematic section.

**92:** "... entailing the need..." entailing is an awkward word choice: perhaps 'pointing to the need' or 'elucidating the need'

The section has been reworded which removed the problematic section.

94: remove the word help before 'improving' and change 'improving' to 'improve'

The section has been moved to the end of the first paragraph and it has been reworded.

**95/96:** you compare to calibrations from the Indian Ocean – are these previously published calibrations? If so list the citations if they are not too lengthy?

The section has been moved to the end of the first paragraph and references have been added.

108: should say... "...core-top transect is comprised of..."

Done.

**108:** you only indicate 2 water masses in this first list... and go on to introduce the third later but it is slightly confusing/misleading to say three here and not list all three

This has been clarified (please see earlier comment by referee 1).

**113:** where are the CTD temperature data from? When were they collected relative to the core-top collections?

This has been clarified.

**134:** no need for brackets around "where possible" when you are using commas

**Has been corrected.**

**144:** NIOP929 – what is this? Which core? From what cruise/expedition/researchers?

The information has been added.

**145:** The sentence should read: "The samples were wet sieved over..."

Has been corrected.

**147:** The sentence should read: "....remove silicates, a hydrogen peroxide treatment to remove organic matter, and followed by an acid ..."

Has been corrected.

149: "depend" should be "depends"

Has been corrected.

**153:** The sentence should read: "...except for a reduction in duration of the methanol washes (25 seconds..."

The section has been reworded which removed the problematic section.

**154:** The language "following previously analysed samples in the laboratory." Is not clear – please clarify and/or rephrase.

This has been clarified.

**159:** for how long was the hydrogen peroxide treatment? At what temperature?

This has been clarified.

**166:** Does this also mean you lose Mg in the glass slide method as well as Ca in order to keep the ratios similar to the pin method this must be true?

The section has been reworded.

**178:** If the data in Figure 4 below show experiment 2 results, then pointing to it here for experiment 1 calcium values doesn't make sense

This has been clarified.

**183:** Did you run a sensitivity test/ leverage test to see how much leverage that outlier has on your regression/model? if it's low then you can likely include it, if it's high then it's likely skewing the results.

We thank the reviewer for this suggestion. We did run a sensitivity test and the  $R^2$  did only slightly change (to roughly 0.9 without the data point with the high Mg/Ca value).

**184:** Is it really true that there is no relationship between Al/Ca and Mg/Ca? or is it just non-linear like logarithmic? Looks like it would be.

We are not confident that the data in figure 4b support a robust definition of a relationship.

**185:** The correlation for not crushed is only really strong if the outlier is included, otherwise just clusters... the correlation for crushed tests is linear however

This section has been reworded.

**224:** Please describe in more detail what the standards were - you should have that information from the lab in which these samples were run and it's standard practice to include the type/name of the specific standards used.

We did specify which standard has been used (ECRM 752-1) and reported statistical data regarding the reproducibility of standards. This passage is contained in the most recent version of the manuscript as well.

**225:** You should rename this Figure B1 - to indicate you are referring to Figure 1 of Appendix B and not Figure 1 of the manuscript.

Has been corrected.

229: 'effect' should be 'effects'

Has been corrected.

**Section headings 3.1 and 3.2** - you refer to more than just Mg/Ca ratios in these sections, perhaps re- naming to say elemental ratios or something similar?

We thank the reviewer for this suggestion. We have changed the headings accordingly.

**261-264:** It does not look like the samples are below the contamination thresholds? you even indicate in table 2 which samples are eliminated due to contamination so this statement can't be true.

We have corrected an error in figure 5 and checked the wording of the respective passage.

**298:**  $r^2$  is a different metric from  $R^2$  – you have used both throughout the text, so go through and ensure

We have checked this and added corrections where needed.

**310:** Perhaps a sentence following indicating how many samples remained for the core-top calibration is useful here.

We thank the reviewer for this suggestion. Upon reading this section again, we feel that the text combined with the table (which are being referred to) provides the information needed for the reader.

**318:** capitalize the 'f' in figure 9 – Figure 9

Has been corrected.

**323:** This calibration is based on 4 samples, one of which is quite far away from the others could be contributing significant leverage to the calibration i.e. without it the entire relationship would fall away. Have you tested this? Also, if you are going to report this calibration model (or any throughout the manuscript) it is good practice to indicate the number of samples/data points you have that built the model. n this case: n = 4.

We have indicated the number of data points in the revised version of the text.

**348:** Maybe rather than 'abnormally' say anomalously high, not abnormally since you have cited evidence where similar reports have been made

Done.

**350-354:** So did you exclude this sample from calibration models? make it very clear if so.

Given that most C. mundulus samples showed indications of remanent contamination, no robust calibration model could be established. The regression models in figures 9 and 11 are indicated as being tentative. This includes a specification that the C. mundulus sample with the high Mg/Ca ratio has been included in the model.

**355- 360:** So did you exclude this sample from calibration models? make it very clear if so.

We have clarified this.

**362:** Are these studies also reporting only c. wuellerstorfi calibration models or which species are they reporting? indicate this here if they are species-specific or mixed-species models!

These studies also report C. wuellerstorfi-only calibration models.

**369:** you have ??? after Figure in your reference to figure 11

Has been corrected.

**393:** the word maybe should be two words in this context: may be

Has been corrected.

**426:** ) close (Figure 6) with a bracket

Has been corrected.

**428:** under what subsection/subtitle number are you pointing readers to for discussion on degree of contamination?

This has been clarified in the new manuscript.

**474:** This first sentence is awkward wording...do you mean to say your core top samples are closely located to previously published data?

The section has been reworded which removed the problematic section.

**514:** risk should be risks

Has been corrected.

**517-518:** How? if you are to suggest more work should be done it's great if you can point to examples of next steps

We thank the reviewer for this suggestion. We have added specific suggestions.

**520:** benthic/planktic should be benthic versus planktic

The section has been reworded.

**526-527:** Also include that this calibration only includes 4 samples.

Done.

**534-535:** It would be great to say something more about your study, while you are not able to speak on the other uncertainties/limiting factors, you have shown here that cleaning procedures may need to be refined and conducted species-by-species.

We agree that our study suggests that species-by-species cleaning protocols may be needed. We do feel though the last paragraph of the "Summary and Conclusion" chapter does state just this.

**RC3**

In "Western Indian Ocean bottom water temperature calibration — are benthic foraminifera Mg/Ca ratios a reliable palaeothermometry proxy?" Larsson & Jung measure Mg/Ca ratios in benthic foraminifers across a depth / space transect in the Western Indian Ocean to determine if a local Mg/Ca-temperature calibration is appropriate. They also test several cleaning techniques to determine their effects on contamination, sample yield, etc.

I'm no expert in Mg/Ca paleothermometry, but it seems like the half of the paper devoted to cleaning procedures is thorough and useful (if a little hard to follow). The calibration, however, is based on very few points (and anchored by one high-temperature sample), which the authors acknowledge, and is not likely to be used on its own. I think the exercise is still

valuable, given that most of their data fall in the Mg/Ca-temperature space in prior calibrations, but the manuscript needs a clearer through-line, e.g., "best" cleaning protocol established -> despite efforts, majority of the specimens contaminated -> resulting calibration is sparse, but (most of) the data seem reasonable and species- / habitat-specific contamination thresholds, calibrations, etc. are recommended.

I believe that this can be of use to the paleoceanographic community pending major revisions in terms of structure, organization, and clarity — I would also highly recommend a thorough grammatical overhaul, there are numerous minor issues only some of which I've noted below. I don't believe further laboratory analyses are required, although there are one or two instances where they might be helpful.

Thank you for your valuable input and for taking the time to thoroughly review our paper.

Your input on organization and structure has been taken into account. Our methodological approach was aimed at improving measurability of our samples. Despite some persistent contamination, some of the results agree with previous studies (as is indicated by referee 3). However, with regard to the cleaning methods, we have not tested a wide range of approaches. We have followed one established cleaning procedure and adjusted the timings to accommodate for potentially more/less contamination in samples. The reworded version of the text should be clearer in this regard.

**General comments:**

The title is rather vague and doesn't reveal much about the study's true findings. Maybe something like "Persistent contamination issues preclude a simple benthic Mg/Catemperature calibration in the Western Indian Ocean?" I'm sure you can come up with something better.

Thanks for this useful suggestion. The next version will have an improved title.

Your tests of the various cleaning procedure parameters are a major part of the paper, and I would mention it in the title. I think you need to emphasize that this is a valuable contribution to Mg/Ca thermometry, however — it reads to me like a sidenote compared to the calibration until you reach the later part of the manuscript.

If I missed this, I apologize, but any ideas as to why the other Indian Ocean calibrations didn't have as extensive of contamination issues?

The tests of the various cleaning procedure parameters are indeed a major part of the paper however, as the methodology that is used is closely following the methodology developed by Barker et al. 2003, the main point of the paper is to highlight how the methodology by Barker et al. 2003 is adopted to different samples and that more research is needed to explore what adaptations are needed for different samples, it also proves that calibrations comparison might be more difficult if different adaptations have been made in response to varying contamination.

In relation to published work using Mg/Ca data in benthic foraminifera, in the absence of comments on adaptations to Barker et al's (2003) methodology we have assumed it has followed exactly the procedure of Barker et al., 2003. We don't know why previous papers have not run into these issues of contamination, as the cleaning methodology, especially applied to benthic foraminifera is generally known to be more difficult and suggested to require more rigorous cleaning. One possible reason is that specimens from the Indian Ocean calibrations have had significantly lower contamination prior to

the cleaning methodology. Another explanation is if the foraminifera tests from the wider Indian Ocean have been deposited in sediments of lower silicate.

Overall, the figures are well-made and easy to understand. I'm unconvinced this is a good idea, but if you (very lightly) shaded the "contaminated zone" above the peach-colored line e.g., in Figure 6, would it help drive home that almost everything's contaminated, or would it just add clutter?

We have improved the color coding in figure 6 which should increase clarity regarding samples being contaminated (or not).

Line by line comments:

We thank the reviewer for the very thorough assessment of our manuscript. Our responses are listed below.

Lines 24-31: This paragraph seems out of place; reading it, I thought this paper was about to go in a very different direction. I think you could start from line 32 and be fine.

We have removed this paragraph.

Lines 94-97: This whole paragraph or a statement of this kind belongs further towards the beginning – maybe at the end of Section 1?

We thank the reviewer for this suggestion. The passage has been moved to the end of section 1 of the Introduction and it has been reworded to improve clariety of the messaging.

Line 121 (?): What is the small inset panel on the right? I assume it's ship tracks but the information would be good to have in the caption.

Thanks for this comment. The inset panel displays the positioning of the T and S profiles within the GLODAPv2 2023 database. We have clarified this in the manuscript.

Lines 153-216: Can you divide Experiments 1-3 into their own subheadings? I.e., "2.2.1. Preparation experiment 1: XYZ?" As it is now, it's a massive section that's difficult to follow. I could also see this being divided up where you explain the experiments simply and clearly in Section 2.2 and describe your findings in the Results.

We have reworded this section of the text, better emphasizing which data are being referred to in each paragraph. This will hopefully have added the clarity entailed in the reviewers comments.

Line 219: There's inconsistency in foraminifera abbreviations: G. ruber vs. Cibicidoides wuellerstorfi, for example. I would go with the abbreviation, but be consistent either way.

We have corrected this throughout the manuscript.

Line 265: Where's panel D?

Thanks for spotting this error. Has been corrected.

Line 325: Capitalize "c" in "c. wuellerstorfi."

Has been corrected in figure 9.

Line 369: "Figure ???11"

Has been corrected.

Section 4.2. header: Not sure why this is blue?

Thanks for spotting this. Has been corrected.

Lines 467-472: Is there any support for this in the literature or is it speculation? I hate to ask, but any possibility of elemental mapping to support?

Thanks for these comments. Elemental scanning is unfortunately beyond the scope of the project, but we will add references supporting the habit statements.

Lines 478-479: Missing parentheses.

Has been corrected.

Lines 484-493: I think Section 4.8 belongs in the introduction – it's critical motivation for your cleaning tests but you don't bring it up until the end.

We politely disagree with the comment that this paragraph is too long. We have, however, reworded this paragraph, hoping that this has improved clarity of the text.

Lines 495-521: This is too long of a block of text, it's dense and hard to follow.

We have reworded this section of the text in order improve readability/accessibility.

Lines 523-535: This is a great conclusion and summary! Bring some of this clarity to the introduction and method explanations.

Thanks for this comment. Large portions of the text have been reworded hopefully achieving the suggested improvement in clarity.

We have reworded this section of the text in order improve readability/accessibility.